# BRAF[V600E] induces reversible mitotic arrest in human melanocytes via microRNA-mediated suppression of AURKB

Andrew S McNeal[1], Rachel L Belote[2], Hanlin Zeng[2], Marcus Urquijo[2], Kendra Barker[2], Rodrigo Torres[1], Meghan Curtin[2], A Hunter Shain[1], Robert HI Andtbacka[2,3], Sheri Holmen[2,3], David H Lum[2], Timothy H McCalmont[1], Matt W VanBrocklin[2,3], Douglas Grossman[2,3], Maria L Wei[1], Ursula E Lang[1], Robert L Judson-Torres[2,3]*

[1]University of California, San Francisco, San Francisco, United States; [2]Huntsman Cancer Inst., Salt Lake City, United States; [3]University of Utah, Salt Lake City, United States

*For correspondence:
Robert.Judson-Torres@hci.utah.edu

**Abstract** Benign melanocytic nevi frequently emerge when an acquired *BRAF*[V600E] mutation triggers unchecked proliferation and subsequent arrest in melanocytes. Recent observations have challenged the role of oncogene-induced senescence in melanocytic nevus formation, necessitating investigations into alternative mechanisms for the establishment and maintenance of proliferation arrest in nevi. We compared the transcriptomes of melanocytes from healthy human skin, nevi, and melanomas arising from nevi and identified a set of microRNAs as highly expressed nevus-enriched transcripts. Two of these microRNAs—MIR211-5p and MIR328-3p—induced mitotic failure, genome duplication, and proliferation arrest in human melanocytes through convergent targeting of AURKB. We demonstrate that *BRAF*[V600E] induces a similar proliferation arrest in primary human melanocytes that is both reversible and conditional. Specifically, *BRAF*[V600E] expression stimulates either arrest or proliferation depending on the differentiation state of the melanocyte. We report genome duplication in human melanocytic nevi, reciprocal expression of AURKB and microRNAs in nevi and melanomas, and rescue of arrested human nevus cells with AURKB expression. Taken together, our data describe an alternative molecular mechanism for melanocytic nevus formation that is congruent with both experimental and clinical observations.

## Introduction

Cutaneous melanoma is a potentially deadly skin cancer arising from the pigment-producing melanocytes of the human epidermis. An activating mutation in the BRAF proto-oncogene (BRAF[V600E]) drives over half of all cutaneous melanomas (*Garnett and Marais, 2004*; *Davies et al., 2002*; *Pollock et al., 2003*). Yet, when a melanocyte acquires a BRAF[V600E] mutation, the cell does not immediately transform to cancer. Instead, it usually undergoes clonal proliferation followed by stable arrest resulting in a benign skin lesion known as a melanocytic nevus or 'mole' (*Pollock et al., 2003*; *Shain et al., 2015*; *Bastian, 2014*; *Dankort et al., 2009*). Despite the continued expression of BRAF[V600E], the majority of melanocytic nevi remain innocuous for the lifespan of the individual, suggesting that nevus cells have robust intrinsic defenses against hyperproliferation. By inducing hyperproliferation and subsequent arrest, the BRAF[V600E] mutation elicits divergent, biphasic phenotypes within a single cell—a poorly understood phenomenon. Characterization of the mechanisms and environmental factors that

**eLife digest** Lots of people have small dark patches on their skin known as moles. Most moles form when individual cells known as melanocytes in the skin acquire a specific genetic mutation in a gene called *BRAF*. This mutation causes the cells to divide rapidly to form the mole. After a while, most moles stop growing and remain harmless for the rest of a person's life.

Melanoma is a type of skin cancer that develops from damaged melanocytes. The same mutation in *BRAF* that is found in moles is also present in half of all cases of melanoma. Unlike in moles, the melanoma-causing mutation makes the melanocytes divide rapidly to form a tumor that keeps on growing indefinitely. It remains unclear why the same genetic mutation in the *BRAF* gene has such different consequences in moles and melanomas.

To address this question, McNeal et al. used genetic approaches to study melanocytes from moles and melanomas. The experiments identified some molecules known as microRNAs that are present at higher levels in moles than in melanomas. Increasing the levels of two of these microRNAs in melanocytes from human skin stopped the cells from growing and dividing by inhibiting a gene called *AURKB*. This suggested that these microRNAs are responsible for halting the growth of moles.

Introducing the mutated form of *BRAF* into melanocytes also stopped cells from growing and dividing by inhibiting *AURKB*. However, changing the environment surrounding the cells reversed this effect and allowed the melanocytes to resume dividing. In this way the mutated form of *BRAF* acts like a switch that allows melanocytes in skin cancers to start growing again under certain conditions.

Further experiments found that a drug called barasertib is able to inhibit the growth of melanoma cells with the mutant form of *BRAF*. Future work will investigate whether it is possible to use this drug and other tools to stop skin cancer tumors from growing, and possibly even prevent skin tumors from forming in the first place.

distinguish the two phenotypes could illuminate candidate strategies for chemoprevention of melanocytic nevus formation or interception of melanoma initiation.

The BRAF$^{V600E}$ mutation is clonally present in ~ 80% of human melanocytic nevi (*Pollock et al., 2003*) and melanocyte-specific expression of the oncogene is sufficient to induce nevus formation in mice (*Dankort et al., 2009*; *Damsky et al., 2015*; *Ruiz-Vega et al., 2020*), suggesting that BRAF$^{V600E}$ drives proliferation arrest in melanocytes. The prevailing theory to explain this seemingly paradoxical role for BRAF$^{V600E}$ is oncogene-induced senescence (OIS) (*Michaloglou et al., 2005*). Cellular senescence is defined as the permanent cell-cycle arrest of previously replication-competent cells (*He and Sharpless, 2017*; *Roy et al., 2020*). Senescence is associated with a variety of molecular hallmarks including elevated expression of p16$^{INK4A}$ and other cyclin-dependent-kinase inhibitors, TP53, H2AX, lysosomal beta-galactosidase and senescence-associated secretory proteins (SASPs) as well as a DNA content profile indicative of arrest in the G0/G1 phases of the cell cycle (*Roy et al., 2020*; *Serrano et al., 1997*; *Campisi and d'Adda di Fagagna, 2007*; *Collado and Serrano, 2006*; *Afshari et al., 1993*; *Chatsirisupachai et al., 2019*). The concept of OIS as a barrier to tumorigenesis derived from observations of a durable proliferation arrest induced by overexpression of oncogenic RAS or RAF in immortalized human fibroblasts (*Serrano et al., 1997*; *Zhu et al., 1998*).

More recent studies have questioned whether the term 'senescence' aptly describes the proliferation arrest of melanocytic nevi. Cell cycle re-entry that accompanies nevus recurrence (*King et al., 2009*), eruption (*Burian and Jemec, 2019*), or transformation to primary melanoma suggests that the nevus arrest phenotype is reversible rather than permanent. Similarly, the expression of senescence markers does not readily distinguish human melanocytic nevi from primary or transformed melanocytes in humans (16$^{INK4A}$, TP53, H2AX, and beta-galactosidase) or mice (any known hallmark of senescence) (*Ruiz-Vega et al., 2020*; *Cotter et al., 2007*; *Tran et al., 2012*). Setting aside the permanence of OIS, melanocytic nevus arrest was largely thought to be driven by induction of p16$^{INK4A}$ expression from the *CDKN2A* gene. Indeed, early studies demonstrated that BRAF$^{V600E}$ induces p16$^{INK4A}$ expression and that melanocytic nevi frequently express high levels of p16$^{INK4A}$ (*Michaloglou et al., 2005*). Despite this, p16$^{INK4A}$ expression is neither ubiquitous across melanocytic nevi, nor uniform within the cells of a single nevus (*Oaxaca et al., 2020*; *Zeng et al., 2018*), and neither knockdown of the 16$^{INK4A}$ transcript nor ablation of the *CDKN2A* locus prevents the onset or rescues BRAF$^{V600E}$-induced

proliferation arrest (*Michaloglou et al., 2005*; *Zeng et al., 2018*; *Haferkamp et al., 2009*; *McNeal et al., 2015*). BRAF$^{V600E}$ also stimulates expression of p15$^{INK4B}$—the translational product of the *CDKN2B* gene (*McNeal et al., 2015*) which is situated adjacent to the *CDKN2A* locus—suggesting BRAF$^{V600E}$ might orchestrate arrest via multiple cyclin-dependent kinase inhibitors that induce arrest in the G1 phase of the cell cycle, prior to genome synthesis. While the loss of the *CDKN2A/B* locus is a formative event in melanoma progression, genetic evolution studies of melanoma progression suggest that selection against the *CDKN2A/B* locus occurs after BRAF$^{V600E}$ melanocytes have already escaped nevus-associated arrest (*Zeng et al., 2018*; *Shain et al., 2018*). Collectively, these data complicate the straightforward model wherein BRAF$^{V600E}$ induces the expression of cell cycle regulators leading to proliferation arrest, which is later circumvented by the genetic loss of those regulators.

Given these observations, we reasoned that the arrest response of human melanocytes to BRAF$^{V600E}$ might be distinct from previous reports of BRAF$^{V600E}$-induced OIS in other cell types. Specifically, we hypothesized that nevus-associated arrest is conditional and coordinated by reversible changes in gene expression. Here, we identify a transcriptional program that is consistently elevated in melanocytic nevi and includes microRNAs (miRNAs) previously shown to have diagnostic value (*Babapoor et al., 2016*; *Torres et al., 2020*). In parallel, we observe that, unlike RAF-induced OIS in fibroblasts, BRAF$^{V600E}$-induced arrest in human melanocytes is reversible and conditional. Merging these two observations, we discover that in melanocytes, regulation of miRNAs coupled to differentiation status permits toggling between BRAF$^{V600E}$-driven hyperproliferative to arrested phenotypes.

## Results

### Melanocytic nevus-enriched miRNAs induce mitotic failure and proliferation arrest in primary human melanocytes

Since melanocytic nevi do not consistently express known senescence markers, we sought to characterize the transcripts that are specifically elevated within nevus melanocytes (*Figure 1A*). We analyzed previously published data sets comparing RNA expression between benign and malignant melanocytic lesions (*Shain et al., 2018*; *Torres et al., 2020*). Each clinical specimen contained matched RNA samples derived from melanocytic nevi and melanomas that arose from those nevi. We reasoned that downregulated transcripts within these newly formed melanomas could represent the most immediate barriers to melanocyte growth. Of the differentially expressed genes (adjusted p value<0.05), 6 of the top 10 with elevated expression in nevi were non-coding miRNAs (*Figure 1B* and *Supplementary file 1*). Among these were miRNAs previously shown as enriched in nevi compared to melanomas, including MIR211-5p, MIR125A-5p, MIR125B-5p, and MIR328-3p (*Torres et al., 2020*; *Latchana et al., 2016*). To determine whether elevated expression of these miRNAs distinguished nevus melanocytes from normal epidermal melanocytes, we profiled melanocytes from fresh healthy skin and nevi (*Supplementary file 2*). Three of the miRNAs (MIR211-5p, MIR328-3p, and MIR125B-5p) were also more highly expressed in melanocytic nevi as compared to healthy melanocytes or melanomas—an expression pattern consistent with enrichment in arrested melanocytes—whereas one miRNA (MIR125A-5p) presented a pattern consistent with lineage-specific expression that is lost upon transformation (*Figure 1C*). We conclude that MIR211-5p, MIR328-3p, and MIR125B-5p are transcripts specifically enriched in arrested melanocytic nevi.

To investigate whether elevated expression of MIR211-5p, MIR328-3p, or MIR125B-5p induce melanocytic arrest, we introduced synthesized mimetics of the mature miRNAs into primary human melanocytes and tracked proliferation over 7 days in culture. The introduction of either MIR211-5p or MIR328-3p resulted in significantly fewer melanocytes compared to a non-targeting control (*Figure 1D*). Increased concentrations of these mimics resulted in a dose-dependent decrease in EdU uptake, consistent with a cell cycle defect (*Figure 1E*). To better characterize the effect of MIR211-5p or MIR328-3p expression on cell cycle and cell viability, we performed time-lapsed quantitative phase imaging (QPI). We and others have previously established QPI as a method for adherent cell cytometry that can simultaneously measure melanocyte proliferation, cell death, and cell cycle stage with single-cell resolution (*Hejna et al., 2017*; *Barker et al., 2020*; *Falck Miniotis et al., 2014*). Images were taken every hour for 5 days post-nucleofection (*Figure 1F*). Cells expressing MIR211-5p or MIR328-3p exhibited significantly reduced growth curves (*Figure 1G*). We observed no evidence of increased cell death (*Figure 1H*), and confirmed this result with Annexin V staining (*Figure 1I*). Experiments

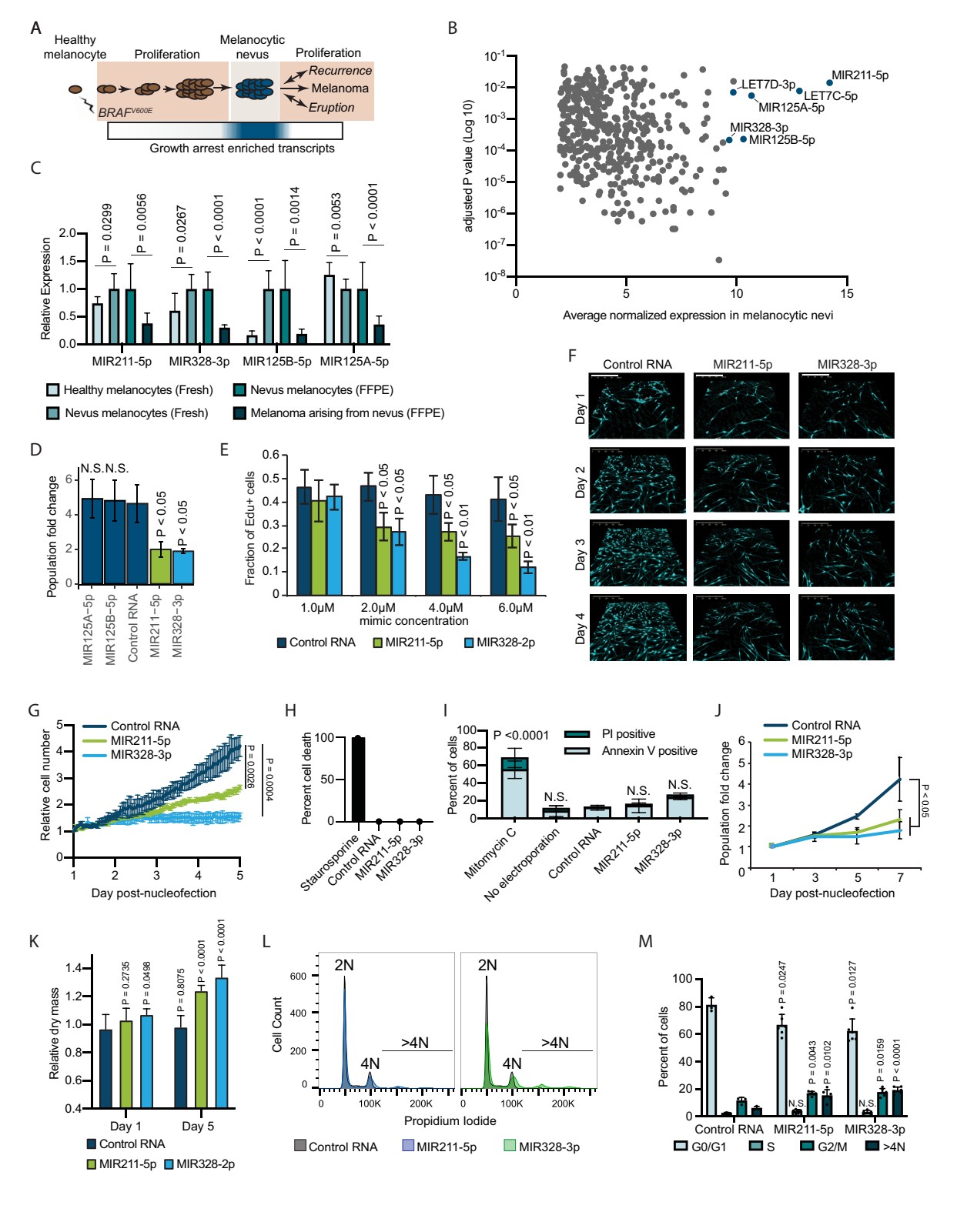

**Figure 1.** Melanocytic nevus enriched miRNAs induce arrest in primary human melanocytes. (**A**) Schematic of hypothesized plastic nature of BRAF$^{V600E}$- and nevus-associated proliferation arrest. Expression of the predicted nevus enriched transcriptional program is highlighted in blue. (**B**) Scatter plot depicting gene expression combining publicly available sequencing data sets totaling 14 melanoma-arising-from-nevus specimens, dissected into benign and malignant portions (28 total samples). Plotted are the adjusted p values (DESeq2) comparing matched nevus and melanoma samples

*Figure 1 continued on next page*

*Figure 1 continued*

against average normalized expression in nevus portion. Only genes exhibiting both high expression and significant differential expression in nevus portions are plotted (see Materials and methods). The full list of genes is included as **Supplementary file 1**. (**C**) Left two columns, the relative expression of indicated miRNAs in freshly isolated human melanocytes from BRAF^V600E melanocytic nevi (n=6) compared to BRAF^WT healthy skin melanocytes (n=8). Right two columns, the relative expression of indicated miRNAs in melanocytes dissected from FFPE nevus (n=7) or melanoma (n=7) specimens. Data represent mean and standard deviation of normalized RNA sequencing read counts relative to nevus-derived samples. (**D**) Mean and standard deviation for population fold change over 7 days of melanocytes nucleofected with indicated miRNA mimics compared to non-targeting control (Control RNA) (n=3). (**E**) Mean and standard deviation for fraction of EdU positive cells 4 days after nucleofection with indicated concentrations of miRNA mimics (n=3). P values indicate comparisons to concentration matched control. (**F**) Representative QPI images of melanocytes nucleofected with indicated miRNA mimics. Pixel color indicates low (black), mid (blue), or high (red) optical density. Scale bar=200 μm. (**G**) Mean and standard deviation for relative QPI-derived cell number over time (n=3). (**H**) Mean and standard deviation for QPI-derived cell death counts as percentage of total cells imaged on day 1 (n=3). Cells treated with 0.1 μM staurosporine were included as a control (n=1) for induction of cell death. (**I**) Mean and standard deviation for percent Annexin V/PI positive cells nucleofected with indicated miRNA mimics (n=3). Cells treated with 5 μg/ml mitomycin C were included as a control (n=3) for induction of apoptosis. (**J**) Mean and standard deviation for relative cell number over time (n=3). (**K**) Mean and standard deviation for relative QPI-derived dry mass per cell (n=3). (**L**) Representative histograms of DNA content profiling via propidium iodide incorporation measured by flow cytometry 5 days post-nucleofection of miRNA mimics. (**M**) Mean and standard deviation for percent of cells in indicated phases of cell cycle based upon profiles as in (**K**) (n=3–6, individual datapoints shown). P values for D-L calculated by unpaired t-tests comparing experimental to Control RNA. N.S, not significant (p≥0.05).

The online version of this article includes the following figure supplement(s) for figure 1:

**Figure supplement 1.** Still frames from **Video 1** documenting representative cytokinesis failure in primary melanocytes transduced with MIR211-5p.

performed using three additional melanocyte preparations counted over 7 days post nucleofection yielded similar results (**Figure 1J**). Thus, MIR211-5p and MIR328-3p induce proliferation arrest in human melanocytes.

Further characterization of the QPI images revealed a distinct increase in intra-cellular dry mass in proliferation-arrested conditions (**Figure 1K**). The dry mass of a cell is a measurement of total biomass (DNA, RNA, proteins, lipids, etc.) and is quantitatively measured by QPI. Relative dry mass fluctuates with cell cycle stage in a predictable manner (**Falck Miniotis et al., 2014**). The 20–40% increase we observed here is characteristic of cells that have doubled their DNA content but failed to complete mitosis. We confirmed this finding by measuring DNA content of the arrested populations (**Figure 1L**). We observed significant increases of cells with both 4 N DNA content and greater than 4 N DNA content, indicative of cytokinesis failure following DNA replication (**Figure 1M**). Consistent with this interpretation, time-lapsed imaging revealed incidences of tripolar mitosis and anaphase bridging in miRNA-expression melanocytes—both processes associated with cytokinesis failure (**Normand and King, 2010**)—which were not observed in control conditions or standard culture (**Video 1** and **Figure 1—figure supplement 1**). We conclude that MIR211-5p and MIR328-3p induce mitotic failure and proliferation arrest when introduced into primary human melanocytes.

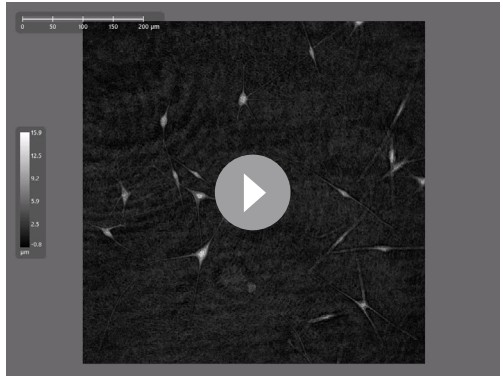

**Video 1.** Cytokinesis failure associated with MIR211-5p expression. Four day time-lapse quantitative phase imaging of cytokinesis failure in primary melanocytes transduced with MIR211-5p.

https://elifesciences.org/articles/70385/figures#video1

## Inhibition of AURKB by MIR211-3p and MIR328-5p restricts proliferation in nevi

To investigate the mechanism by which MIR211-5p and MIR328-3p induce arrest in primary human melanocytes, we applied our previously established pipeline for identification of relevant miRNA targets (**Figure 2A**; **Judson et al., 2013**). First, we nucleofected MIR211-5p, MIR328-3p, or control RNA mimics into primary human melanocytes freshly isolated from four independent donors and conducted RNA sequencing (**Supplementary file 3**). We next selected all genes that were significantly downregulated by either miRNA as compared to the control RNA, and further refined the set to the computationally predicted targets of MIR211-5p and/or MIR328-3p. Finally, we

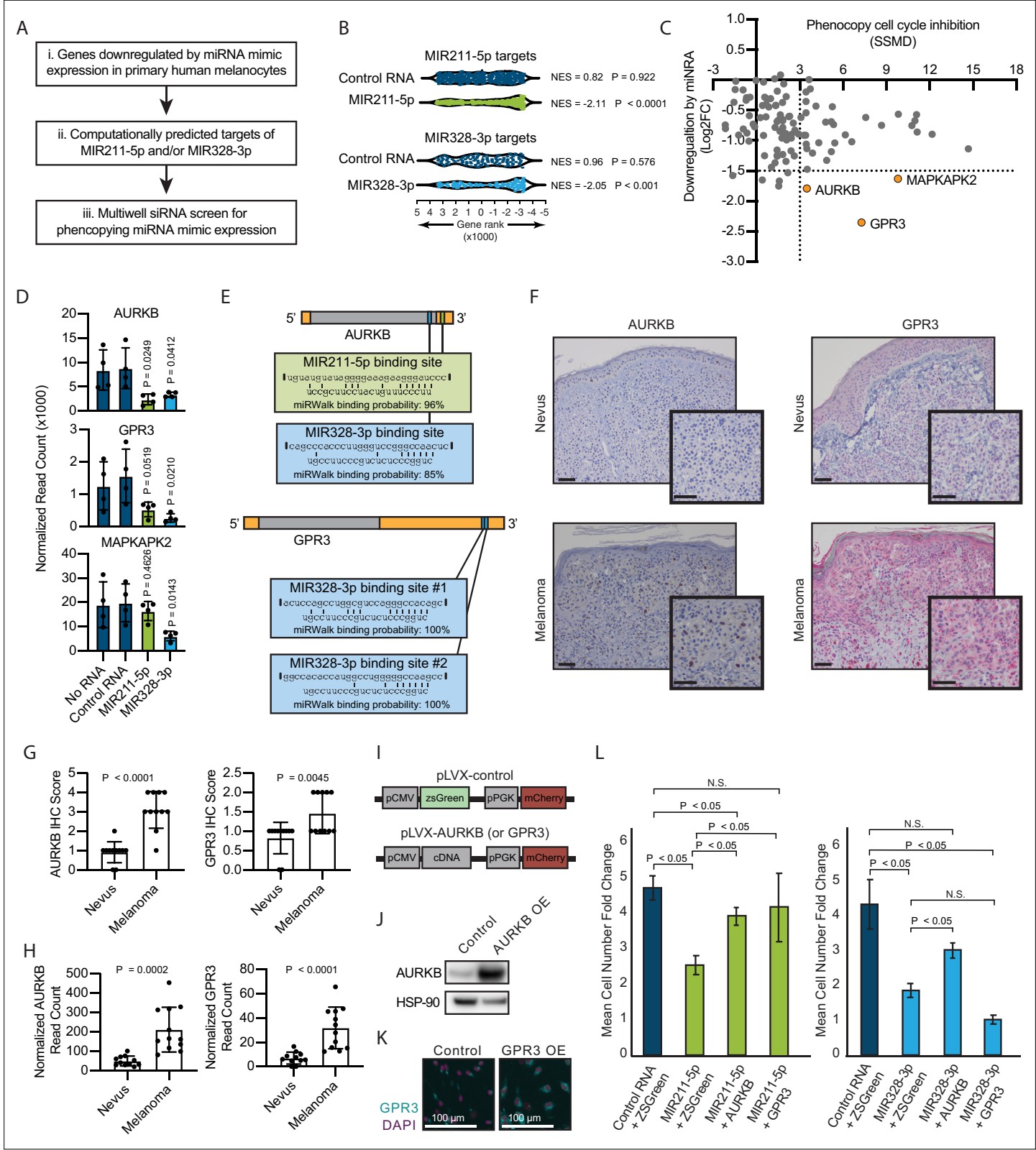

**Figure 2.** Inhibition of AURKB by MIR211-3p and MIR328-5p restricts proliferation in nevi. (**A**) Schematic of experimental and computational pipeline for identifying targets of MIR211-3p and/or MIR328-3p responsible for proliferation arrest. (**B**) GSEA analysis comparing predicted MIR211-5p and MIR328-3p target mRNAs to changes in gene expression after nucleofection with indicated mimics compared to nucleofection control . *Supplementary file 3* provides full differentially expressed gene lists. (**C**) Plot of siRNA screen results for phenocopying proliferation arrest versus log2 fold change in

*Figure 2 continued on next page*

*Figure 2 continued*

expression after miRNA mimic nucleofection. Strictly standardized mean difference (SSMD) reports the magnitude and significance of inhibition of Edu incorporation (n=3, see *Supplementary file 4* for list of genes and screen results). Dotted lines indicate cutoff values for further evaluation. Orange points indicate genes meeting cutoff criteria. (**D**) Mean and standard deviation for expression of indicated genes after nucleofection of indicated mimics (n=4). P values calculated by unpaired t-tests compared to no RNA. (**E**) Predicted binding sites and computed binding probabilities of MIR211-5p and MIR328-3p in AURKB and GPR3 transcripts. (**F**) Representative images of immunohistochemical staining for AURKB (brown chromagen) and GPR3 (red chromagen) expression in FFPE samples of nevi and melanoma. Scale bars=50 μm. (**G**) Mean and standard deviation for immunohistochemical staining scores as in (**F**) for cohorts of 11 nevi and 11 melanomas. P values calculated by unpaired t-tests. (**H**) Mean and standard deviation for read counts for cohorts of 11 nevi and 12 melanomas. P values calculated by unpaired t-tests. (**I**) Design of pLVX vectors expressing either AURKB or GPR3. (**J**) Western blot of AURKB expression in human melanocytes with or without lentiviral AURKB overexpression. HSP90 is the loading control. Source data provided as Supplementary files (*Figure 3—source data 1*). (**K**) Representative photomicrographs ( 20×) of immunofluorescence for GPR3 (green) or (DAPI) (purple) in human melanocytes with GPR3 lentiviral overexpression (GPR3 OE) or without (Control) overexpression. (**L**) Mean and standard deviation for cell number fold change over 7 days after nucleofection with indicated miRNA mimics in melanocytes transduced with lentiviral constructs expressing zsGreen, AURKB, or GPR3 (n=6 melanocyte preps, type 2 t test, N.S., not significant (p≥0.05)). P values calculated by unpaired t-tests. GSEA, gene set enrichment analysis; NES, normalized enrichment score; p, nominal p value.

The online version of this article includes the following figure supplement(s) for figure 2:

**Source data 1.** Zip files containing raw and annotated images of Western blots.

nucleofected siRNAs targeting the 115 resulting genes and assayed for reduced EdU incorporation that would phenocopy MIR211-5p or MIR328-3p expression (*Supplementary file 4*). As expected, computationally predicted targets of MIR211-5p and MIR328-3p were enriched in the set of down-regulated genes following MIR211-5p or MIR328-3p nucleofection, respectively (*Figure 2B*). AURKB, GPR3, and MAPKAPK2 showed the greatest magnitude of repression due to miRNA expression with significant EdU reduction upon siRNA-mediated knockdown (*Figure 2C*). Of these, AURKB (predicted MIR211-5p target) and GPR3 (predicted MIR328-3p target) were significantly inhibited upon nucleofection of either miRNA (*Figure 2D*). We reasoned these genes might represent conver-gent nodes driving proliferation arrest. Further exploration of the AURKB transcript revealed a poten-tial MIR328-3p binding site just before the 3′ untranslated region (*Figure 2E*; *Dweep et al., 2014*). In contrast, we were unable to identify a candidate MIR211-5p binding site in the GPR3 transcript, suggesting indirect inhibition of this gene by MIR211-5p.

To determine whether decreased AURKB and/or GPR3 expression was associated with melanocytic nevi, we acquired 23 clinical specimens and measured both the transcript and protein expression of these genes (*Figure 2F–H*). The expression of both AURKB and GPR3 protein and mRNA was enriched in melanomas as compared to nevi (*Figure 2G–H*). Taken together, these data demonstrate an inverse expression between two miRNAs (MIR211-5p and MIR328-3p) and two targeted mRNAs (AURKB and GPR3) that toggles concurrently with the transition of arrested melanocytic nevi to mela-nomas. To further investigate whether either mRNA was sufficient to rescue miRNA-mediated arrest, primary melanocytes were transduced with constructs expressing either zsGreen (control), AURKB, or GPR3 (*Figure 2I-K*), then subsequently nucleofected with either MIR211-5p or MIR328-3p. Expression of either mRNA partially rescued MIR211-5p-induced arrest, while AURKB, but not GPR3, partially rescued MIR328-3p-induced arrest (*Figure 2L*). Thus, while other MIR211-5p and MIR328-3p targets, including GPR3, may play a role in the induction of melanocyte proliferation arrest, these data suggest that AURKB inhibition is a critical convergent node of both miRNAs in facilitating proliferation arrest in nevi. This interpretation is further supported by the well-established role of AURKB as an essential orchestrator of successful mitosis (*D'Avino and Capalbo, 2015*).

## BRAF$^{V600E}$ induces a reversible and conditional proliferation arrest in human melanocytes

We hypothesized that AURKB inhibition by MIR211-5p and MIR328-3p may be the underlying mech-anism by which BRAF$^{V600E}$ induces proliferation arrest in melanocytes. BRAF$^{V600E}$ expression has been demonstrated to result in an accumulation of growth-arrested fibroblasts with 2 N DNA content, presumably in a permanent G0/G1 senescent state (*Michaloglou et al., 2005*; *Serrano et al., 1997*; *Zhu et al., 1998*). In contrast, upon overexpression of MIR211-5p or MIR328-3p, we observed an accumulation of arrested melanocytes with ≥4 N DNA content, and a significant decrease in cells with 2 N DNA content (*Figure 1M*). To determine the DNA content profile of primary human melanocytes

under BRAF$^{V600E}$-induced arrest, we utilized an established lentiviral construct for dose-responsive doxycycline-inducible BRAF$^{V600E}$ expression (*Figure 3A*; *McNeal et al., 2015*). We monitored melanocyte growth under a titration of doxycycline and identified the lowest concentration (15.6 ng/ml) that induced sustained arrest after 40 hr exposure (*Figure 3B*). Similar to our observations for miRNA-induced arrest, relative per cell dry mass increased by 50% concurrent with BRAF$^{V600E}$ induced-arrest (*Figure 3C*). Cell cycle profiling confirmed a significant and dose-dependent accumulation of cells with ≥4 N DNA content and a decrease in cells with 2 N DNA content (*Figure 3D–E*). The cell cycle profiles were notably dissimilar to that of melanocytes treated with a pharmacologic CDK4/6 inhibitor (PD0332991, Palbociclib) (*Clark et al., 2016*) known to induce G0/G1 arrest. Treatment with a pharmacologic AURKB inhibitor (AZD2811, Barasertib) (*Helfrich et al., 2016*) that induces G2/M arrest and mitotic failure also resulted in an accumulation of melanocytes with ≥4 N DNA content and a decrease in cells with 2 N DNA content.

In addition to the accumulation of 2 N cells, previous reports of RAF expression in fibroblasts show that transient expression can induce a durable arrest phenotype (*Zhu et al., 1998*). To test the durability of BRAF$^{V600E}$-induced arrest in human melanocytes, we induced doxycycline-dependent BRAF$^{V600E}$ expression for 48 hr and monitored cell number for an additional 2 days to confirm arrest. Media was then replaced to remove doxycycline. Within 12 hr of doxycycline removal, the previously arrested melanocytes began to proliferate (*Figure 3F*), demonstrating that BRAF$^{V600E}$-induced proliferation arrest in melanocytes is reversible and dependent on continued oncogene expression.

Nevi recurrence after excision (*King et al., 2009*) and eruption upon external stimuli (*Burian and Jemec, 2019*), together with our observation that BRAF$^{V600E}$ induced arrest is reversible in melanocytes, suggested that BRAF$^{V600E}$-induced arrest may depend on secondary signals from the microenvironment. Tetradecanoylphorbol acetate (TPA) is a protein kinase C (PKC) activator commonly used as a mitogen in primary melanocyte media to substitute for endothelin receptor type B activation (*Hsu et al., 2005*). In contrast, TPA also inhibits the cell cycle of melanoma cells (*Coppock et al., 1995*; *Arita et al., 1998*). BRAF$^{V600E}$ expression permits growth of the Melan-A cell line in the absence of TPA (*Wellbrock et al., 2004*). We therefore wanted to determine the relative contributions of TPA and BRAF$^{V600E}$ to the BRAF$^{V600E}$-induced primary melanocyte arrest in vitro. We induced BRAF$^{V600E}$ expression in primary human melanocytes in two medias—one containing TPA (+TPA) or one containing the endothelin receptor type B ligand, endothelin-1 (TPA-free) and observed divergent dose-responsive phenotypes (*Figure 3G*). In the presence of TPA, BRAF$^{V600E}$ expression inhibited cell proliferation. By contrast, in TPA-free media, BRAF$^{V600E}$ expression induced proliferation, a phenotype that persisted for the remaining duration of the primary cultures (4–6 weeks). We next subjected melanocytes to the arrest-inducing conditions of doxycycline treatment in +TPA media for 4 days, then replaced with doxycycline-containing, TPA-free media. We observed a rapid shift to the proliferative phenotype even in the continued presence of doxycycline (*Figure 3H*). To determine whether the observed reversibility of BRAF$^{V600E}$-induced growth arrest was due to insufficient duration or degree of expression of the oncogene, we exposed melanocytes to a high concentration of doxycycline (60 ng/ml) for 30 days prior to switching medias (*Figure 3I*). Melanocytes in +TPA media remained arrested for the duration of doxycycline treatment then grew again upon removal of either doxycycline or TPA (*Figure 3J–K*). From these results, we conclude that BRAF$^{V600E}$ expression induces divergent and interconvertible phenotypes—either hyperproliferation or a reversible arrest—in human melanocytes conditional on extrinsic stimuli.

## BRAF$^{V600E}$-induced arrest is dependent on melanocyte growth conditions and differentiation state

To investigate the relative contributions of BRAF$^{V600E}$ and TPA to the expression of MIR211-5p and MIR328-3p, we conducted small RNA sequencing of melanocytes grown in +TPA or TPA-free media and with or without BRAF$^{V600E}$ expression. The expression of MIR328-3p was relatively stable, demonstrating less than twofold changes across conditions (*Figure 4A–B*, *Supplementary file 5*). In contrast, MIR211-5p was one of the most differentially expressed genes in +TPA conditions compared to TPA-free conditions (adj. p=6.8E−23) (*Figure 4A–B*, *Supplementary file 5*). Induction of BRAF$^{V600E}$ in +TPA conditions did not further alter expression of either miRNA. Direct comparison of the arrested melanocytes in +TPA + Dox conditions to the proliferative melanocytes in −TPA +Dox conditions revealed MIR211-5p as the most significantly downregulated miRNA in the proliferative condition (*Figure 4B*).

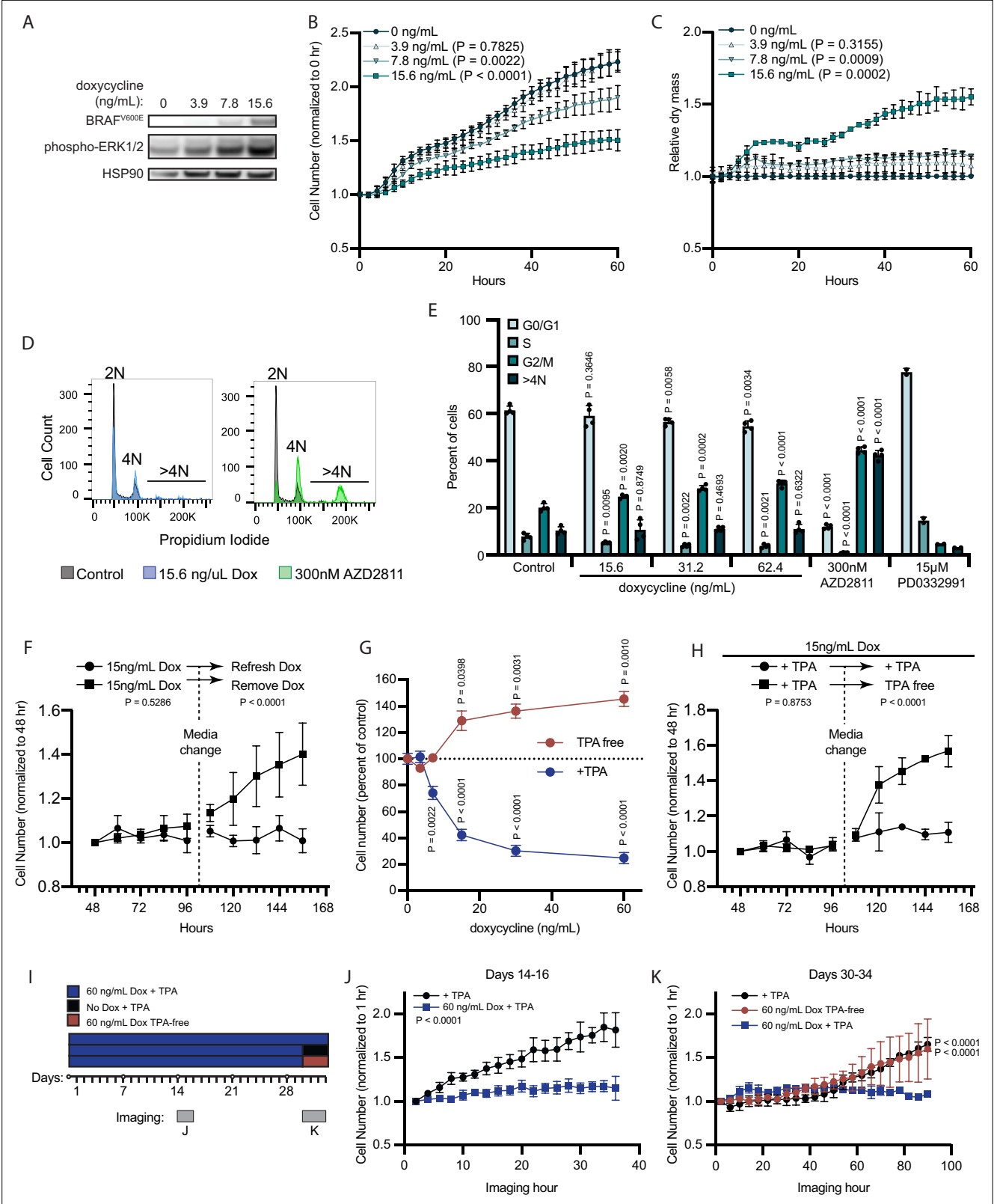

**Figure 3.** BRAF^V600E induces a reversible and conditional arrest in human melanocytes. (**A**) Representative (of n=3) Western blot analysis of BRAF^V600E, phospho-ERK1/2, and HSP90 (loading control) in diBRAF^V600E melanocytes in response to increasing concentration of doxycycline. Source data provided in Supplementary files (*Figure 3—source data 1*). (**B**) Mean and standard deviation for relative QPI-derived cell number at indicated hours of diBRAF^V600E melanocytes in response to increasing concentration of doxycycline (n=6). P values from unpaired t-tests compared to 0 ng/ml condition.

*Figure 3 continued on next page*

*Figure 3 continued*

(**C**) Mean and standard deviation for relative QPI-derived dry mass per cell of diBRAF$^{V600E}$ melanocytes in response to increasing concentration of doxycycline at indicated hours (n=6). P values from unpaired t-tests compared to 0 ng/ml condition. (**D**) Representative histograms of DNA content profiling via propidium iodide incorporation measured by flow cytometry 5 days after treatment with indicated molecules. AZD2811 included as a control for induced cytokinesis failure. (**E**) Mean and standard deviation for percent of cells in indicated phases of cell cycle after treatment with doxycycline based upon profiles as in (**D**) (n=4). AZD2811 (n=4) is a control for induced cytokinesis failure. PD0332991 (n=2) is a control for G0/G1 arrest. P values from unpaired t-tests compared to Control (no treatment) condition. (**F**) Mean and standard deviation for relative QPI-derived cell number of diBRAF$^{V600E}$ melanocytes at indicated hours after treatment doxycycline. Dotted line indicates change in media to either refresh doxycycline or add media without doxycycline (n=3). P values from unpaired t-tests comparing final timepoints. (**G**) Mean and standard deviation for cell number as percent of control of diBRAF$^{V600E}$ melanocytes cultured with or without TPA after 5 days of treatment with indicated concentrations of doxycycline. P values from unpaired t-tests compared to 0 ng/ml conditions. Baseline population doubling times were 3.2 days (+TPA) and 4.2 days (TPA-free). (**H**) Mean and standard deviation for relative QPI-derived cell number of diBRAF$^{V600E}$ melanocytes at indicated hours after treatment doxycycline. Dotted line indicates change in media to either refresh TPA-containing media or add media without TPA (n=3). P values from unpaired t-tests comparing final timepoints. (**I**) Schematic of experimental design testing the reversibility of growth arrest after 30 days exposure to high doxycycline (60 ng/ml). Days of live imaging represented in (**J**, **K**) are indicated by gray bars. (**J**) Mean and standard deviation for relative QPI-derived cell number of diBRAF$^{V600E}$ melanocytes at 14 days after addition of doxycycline (n=3). P values from unpaired t-tests comparing final timepoints. (**K**) Mean and standard deviation for relative QPI-derived cell number of diBRAF$^{V600E}$ melanocytes at 30 days after addition of doxycycline (n=3). Media was changed just prior to imaging to either refresh doxycycline (blue), change to +TPA media without doxycycline (black), or change to TPA-free media with doxycycline (red). P values from unpaired t-tests comparing final time points. QPI, quantitative phase imaging; TPA, tetradecanoylphorbol acetate.

The online version of this article includes the following figure supplement(s) for figure 3:

**Source data 1.** Zip files containing raw and annotated images of Western blots.

Thus, in comparison to TPA, BRAF$^{V600E}$ induction had minimal effect on the expression of the miRNAs. The most notable effect was a decrease of MIR328-3p expression coinciding with the induction of BRAF$^{V600E}$ in TPA-free conditions (*Figure 4A*). While MIR328-3p downregulation likely contributes to increased proliferation in this condition, the more pronounced effect of TPA on MIR211-5p expression suggests that the mechanisms governing whether BRAF$^{V600E}$ induces arrest or proliferation are a function of the cell's transcriptional state rather than directly downstream of BRAF$^{V600E}$ itself.

To better characterize the two transcriptional states influencing the BRAF$^{V600E}$ phenotype, we conducted mRNA sequencing of the same specimens. Principal component (PC) analysis revealed that 97.4 % of the variance between samples was explained by two PCs—PC1 ( 81.75 variance) separated transcriptomes based upon exposure to TPA and PC2 ( 15.7% variance) separated transcriptomes based upon BRAF$^{V600E}$ expression (*Figure 4C*, *Supplementary file 5*). We conducted differential expression (DE) analyses to compare all samples exposed to +TPA versus TPA-free media, regardless of BRAF$^{V600E}$ expression (*Figure 4D*, top) and to compare all samples expressing BRAF$^{V600E}$, regardless of TPA exposure (*Figure 4D*, bottom). As expected from the PC analysis and similar to the miRNA profiles, most gene expression variance was due to media conditions with a smaller, but consistent, set of genes activated by BRAF$^{V600E}$.

One initially surprising result of this analysis was the static expression of both MIR211-5p and MIR328-3p in +TPA conditions, regardless of BRAF$^{V600E}$ expression. However, as inhibitors of translation, miRNA function depends on the expression level of its targets (*Mukherji et al., 2011*; *Ovando-Vázquez et al., 2016*), effectively functioning as buffers of gene expression that prevent aberrantly high levels of a transcript (*Ebert and Sharp, 2012*). BRAF$^{V600E}$ has been previously demonstrated as an upstream activator of AURKB transcription in melanoma cells (*Sharma et al., 2013*). We therefore reasoned that MIR211-5p expression in +TPA conditions might dampen AURKB activation by BRAF$^{V600E}$. Consistent with this hypothesis, the addition of doxycycline induced AURKB expression in TPA-free/MIR211-5p low conditions, but this effect was not observed in +TPA/MIR211-5p high conditions (*Figure 4E*).

We next conducted gene set enrichment analyses (GSEA) comparing TPA-regulated genes and BRAF$^{V600E}$- regulated genes to Molecular Signature Database Hallmark gene sets (*Liberzon et al., 2015*), the KEGG PATHWAY database (*Kanehisa et al., 2016*), and signatures important in BRAF$^{V600E}$ signaling (*Ryu et al., 2011*), human melanocyte differentiation (*Belote et al., 2021*), or melanoma cell state (*Hoek and Goding, 2010*; *Tirosh et al., 2016*; *Tsoi et al., 2018*). Among the pathways uniquely enriched with BRAF$^{V600E}$ expression were the expected increases in signatures downstream of BRAF$^{V600E}$, MAPK (KRAS) signaling, and glycolysis (*Haq et al., 2014* ; *Figure 4F* and *Supplementary file 6*). Gene sets uniquely enriched in +TPA media included signatures associated with cell cycle

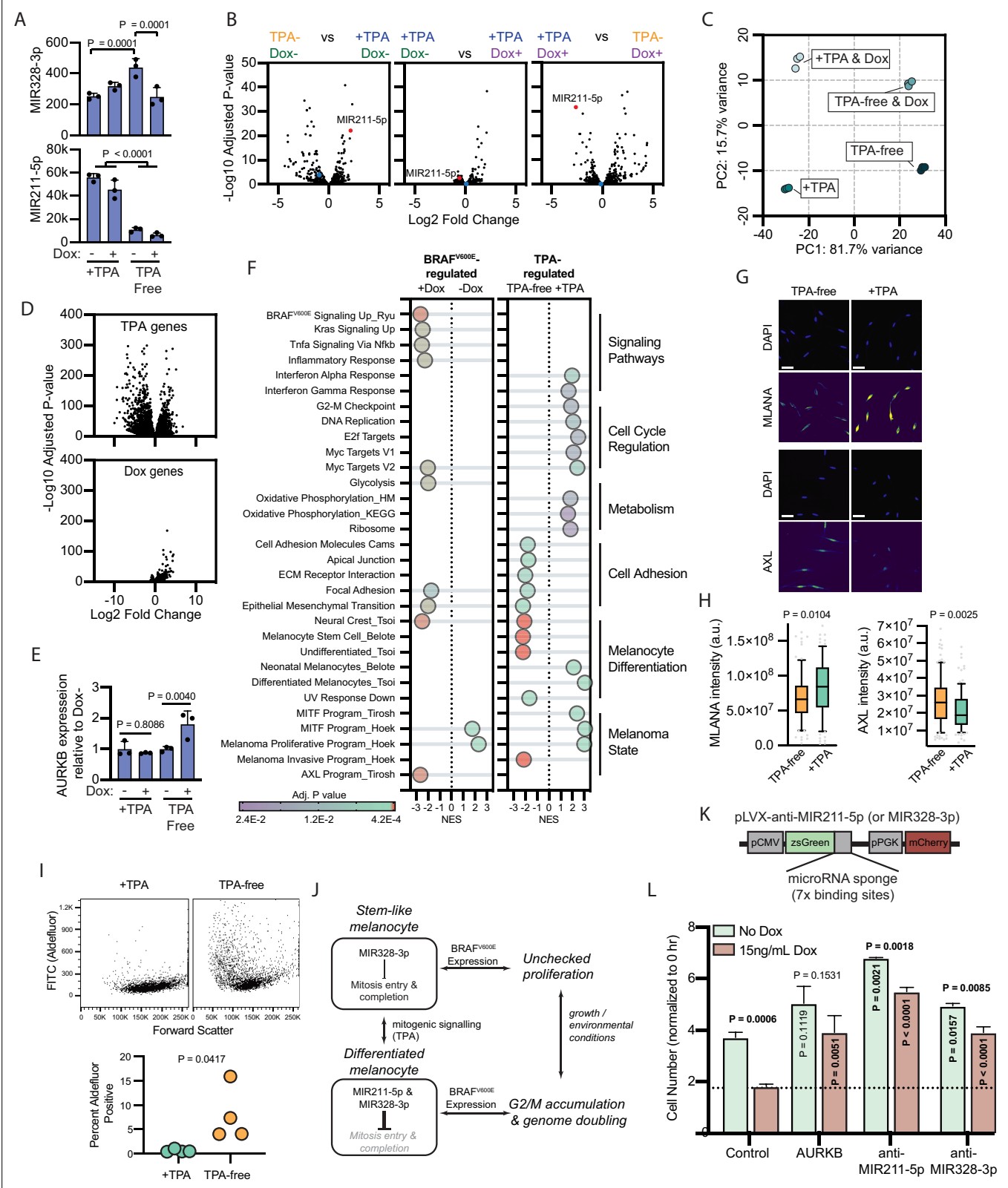

**Figure 4.** BRAF^V600E-induced arrest is dependent on melanocyte growth conditions and differentiation state. (**A**) Normalized read counts for MIR328-3p (top) and MIR211-5p (bottom) in TPA-containing (+TPA) or TPA-free media, with or without 15.6 ng/ml doxycycline (Dox). See **Supplementary file 5** for individual values and exact p values. (**B**) Volcano plots depicting log2 fold change against adjusted p value (−log10) from differential expression (DE) analysis of small RNA sequencing of indicated comparisons. MIR211-5p labeled and shown in red. MIR328-3p shown in blue. (**C**) First and second

*Figure 4 continued on next page*

*Figure 4 continued*

principal components (PCs) separating diBRAF$^{V600E}$ melanocytes cultured in indicated conditions. (**D**) Volcano plots depicting log2 fold change against adjusted p value (−log10) from DE analysis of mRNA sequencing of all +TPA specimens versus all TPA-free specimens (top) or all Dox specimens versus all no-Dox specimens (bottom). (**E**) Normalized read counts for AURKB normalized to no Dox conditions (n=3). P values from unpaired t-tests. Sequencing data represented in (**A–F**) are from n=3 per condition analyzed with DESeq2. (**F**) Gene set enrichment analysis (GSEA) comparing DE gene sets from (**D**) to Molecular Signature Database Hallmark gene sets, KEGG Pathway gene sets and published signatures of melanocyte signaling and differentiation from *Belote et al., 2021*; *Tsoi et al., 2018*; *Tirosh et al., 2016*; *Ryu et al., 2011*; *Hoek and Goding, 2010*. All enrichments with adjusted p value<0.025 are depicted. See *Supplementary file 6* for all associations. Normalized enrichment score (NES). Enrichment differences between pathways across the four conditions in (**C**) are depicted in *Figure 4—figure supplement 1*. (**G**) Representative photomicrographs of immunofluorescence for MLANA or AXL in human melanocytes cultured in +TPA or TPA-free conditions. White scale bar=30 μm. (**H**) Quantification of fluorescence intensity (arbitrary pixel intensity units, a.u.) from experiments represented in (**G**). Box (median, 25th and 75th percentiles) and whiskers (10th and 90th percentiles) of 84, 88, 113, and 96 cells, respectively. P values from unpaired t-tests. (**I**) Representative plots (top) and quantification (bottom, n=4) of relative aldehyde dehydrogenase activity of melanocytes grown in +TPA or TPA-free conditions. Normalized read counts for AURKB normalized to no Dox conditions (n=3). P values from unpaired t-tests. Sequencing data represented in (**A–G**) from n=3 per condition analyzed with DESeq2. (**J**) Schematic of model supported by transcriptomic analyses. TPA induces a more differentiated state in human melanocytes. Due to increased MIR211-5p and base-line MIR328-3p expression, AURKB expression is capped, resulting in G2/M accumulation upon BRAF$^{V600E}$ expression. (**K**) Design of pLVX vectors expressing microRNA sponges fused to zsGreen. (**L**) Mean and standard deviation for cell number as percent of time point 0 hr after 7 days of culture in +TPA media with or without 15.6 ng/ml Dox (n=3 per condition). Melanocytes were transduced with either pLVX-control, -AURKB, -anti-MIR211-5p, or -anti-MIR328-3p. P values from unpaired t-tests compared to matched no Dox condition (horizontal), Control no Dox condition (vertical in green bar), or Control Dox condition (vertical in red bar). TPA, tetradecanoylphorbol acetate.

The online version of this article includes the following figure supplement(s) for figure 4:

**Figure supplement 1.** Summary of gene set enrichment for gene sets in *Figure 1F* across the four conditions in *Figure 4C*.

regulation, such as the G2-M checkpoint, oxidative phosphorylation, the MITF program, and differentiated melanocytes. In contrast, TPA-free media-induced gene sets are associated with cell adhesion and melanocyte progenitor and stem cells. Corroborating this observation, melanocytes grown in TPA-free media expressed higher levels of AXL (*Figure 4G–H*), lower levels of MLANA, a melanocyte differentiation marker, and presented greater activity of aldehyde dehydrogenase, enzymatic activity associated with stemness (*Figure 4I*). These results indicate that exposure to TPA induces a more differentiated melanocyte phenotype, consistent with previous reports (*Prince et al., 2003*; *Vidács et al., 2021*; *Kormos et al., 2011*).

To look for synergist effects of both BRAF$^{V600E}$ expression and media conditions, we performed GSEA analyses on each set of pair-wise conditions (*Supplementary file 6* and summarized in *Figure 4—figure supplement 1*). Generally, most gene sets identified in *Figure 4F* were regulated by only one of the two stimuli. For example, genes downstream of BRAF$^{V600E}$ signaling were only influenced by doxycycline exposure. Interesting exceptions to this trend were the inflammatory response, which required both stimuli; cell cycle regulation gene sets, which were activated by TPA alone but further elevated when supplemented with doxycycline; and epithelial to mesenchymal transition genes, which were activated by either stimulus. A subset of melanocyte dedifferentiation gene sets, specifically neural crest genes, the melanoma invasion program, and the AXL program, but not markers of melanocyte stem cells, were activated by either TPA-free media or BRAF$^{V600E}$ expression, albeit the activation was more pronounced in the former. This observation suggests that BRAF$^{V600E}$ expression alone does induce some programs associated with melanocyte dedifferentiation, but with the retention of MIR211-5p expression, the induction is insufficient to overcome growth arrest.

Collectively, our transcriptomic analyses support a model whereby growth in +TPA conditions induces a more differentiated state in melanocytes coinciding with an increase in MIR211-5p expression (*Figure 4J*). This increased MIR211-5p combined with relatively stable MIR328-3p expression, dampens entry into mitosis and/or successful completion of mitosis at least in part through AURKB inhibition. The attenuation of cell division results in an accumulation of cells in G2/M—an effect that is more pronounced in conditions stimulating accelerated growth, such as with BRAF$^{V600E}$ expression. To test this model, we transduced doxycycline-inducible BRAF$^{V600E}$ melanocytes in +TPA conditions with lentiviral constructs constitutively expressing either AURKB (*Figure 2I–J*) or miRNA sponges (*Figure 4K*) that inhibit either MIR211-5p or MIR328-3p. AURKB expression had no baseline effect on melanocyte growth, but rescued BRAF$^{V600E}$-induced arrest (*Figure 4L*), suggesting the inability of the oncogene to induce AURKB in this context serves as a bottleneck in cell division. Also consistent with the model, sponges against either MIR211-5p or MIR328-3p increased baseline division rate and

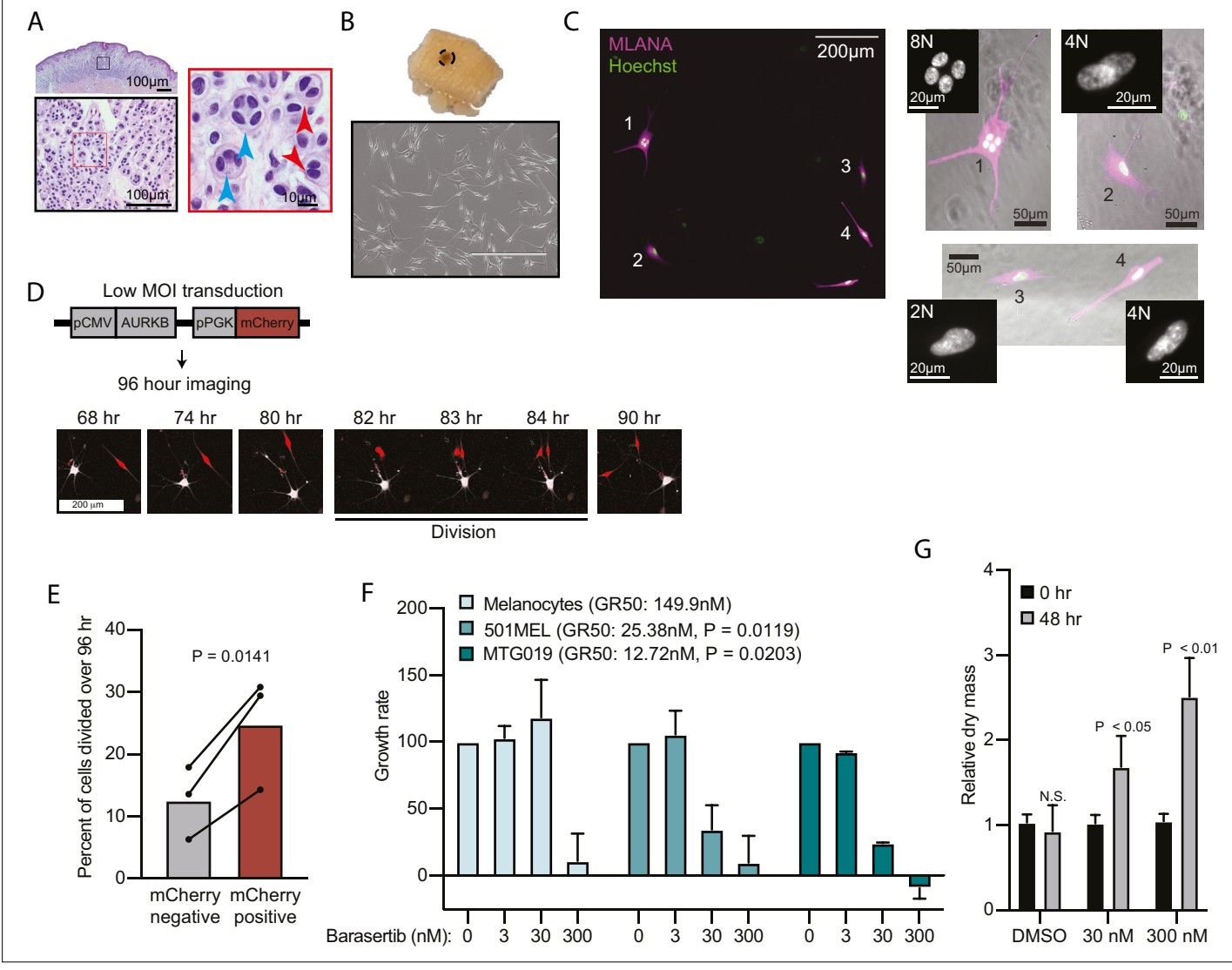

**Figure 5.** AURKB inhibition is critical for BRAF$^{V600E}$-induced arrest of human nevi. (**A**) Representative image of 15 H&E-stained melanocytic nevus. Border colors indicate two consecutive magnifications. Arrowheads indicate bi- (red) or multi- (blue) nucleation. (**B**) Example images of skin specimen containing a melanocytic nevus (circled) and phase-contrast microscopy of melanocytes isolated from nevus portion. Scale bar=400 µm. (**C**) Images of MLANA (purple) and Hoechst (green) co-staining of melanocytic nevus derived melanocytes in culture. Zoomed images show representative 2 N, 4 N, and 8 N cells. (**D**) Representative images of QPI coupled with fluorescence microscopy to identify adjacent mCherry-expressing and mCherry-negative melanocytes. Individual cells were identified at time zero, tracked, and monitored for division for 96 hr. (**E**) Mean and individual matched data points for the percent of n=3 mCherry-negative or mCherry-positive cells identified at time zero that divided over 96 hr. P value from paired t-test. (**F**) Mean and standard deviation for 48 hr growth rates of indicated cells treated with Barasertib. P values from unpaired t-tests comparing the 30 nM samples of melanoma lines to primary melanocytes (n=3). (**G**) Mean and standard deviation for relative QPI-derived dry mass per 501MEL cell at initial time point (black) compared to 48 hr treatment with Barasertib (gray) (n = 3). P values from unpaired t-tests. QPI, quantitative phase imaging.

rescued BRAF$^{V600E}$-induced arrest, demonstrating that both miRNAs contribute to inhibition of melanocyte division, with effects that are more pronounced in the presence of growth signals.

## AURKB inhibition drives melanocytic nevus-associated proliferation arrest

To further investigate whether this mechanism of arrest occurs in clinical nevus specimens, we examined an independent cohort of 15 nevi for evidence of mitotic failure. Histopathologic analysis revealed that all nevi within the cohort exhibited binucleation (*Figure 5A*). The percent observed binucleated melanocytes ranged from 3% to 12% (mean=6.3%). It is important to note that since binucleation can

only be observed in the fraction of cells sectioned perpendicular to the plane of the adjacent nuclei, 3–12% is likely an underestimate. We also observed giant multinucleated cells in 10 of the 15 nevi, indicative of multiple rounds of DNA synthesis and cytokinetic failure and consistent with previous reports (*Rogers et al., 2016*).

We sought to determine whether AURKB inhibition was necessary and sufficient for the multi-nucleated arrest we observed in melanocytic nevi. We obtained three fresh skin biopsies containing nevi, microdissected the nevus portion, and isolated melanocytes (*Figure 5B*). Nevus-derived mela-nocytes appeared healthy in culture. Consistent with the histopathologic cohort, co-staining with melanocyte marker, MLANA, and Hoechst revealed frequent 4 N and 8 N melanocytes (*Figure 5C*). Transduction of nevus melanocytes with an AURKB and mCherry expressing lentiviral vector at low multiplicity of infection showed that the mCherry positive cells divided significantly more than their mCherry negative counterparts (*Figure 5D–E*) in cells from all three nevus donors. This result suggests that expression of AURKB is also rate limiting for the proliferation of human nevus melanocytes, as we observed in primary melanocytes expressing BRAF$^{V600E}$.

To test the sufficiency of AURKB inhibition to limit BRAF$^{V600E}$ hyperproliferation in melanoma, we assessed normalized growth rate inhibition of three cultures—primary human melanocytes, an established BRAF$^{V600E}$ human melanoma cell line, and a short-term BRAF$^{V600E}$ human melanoma culture—when exposed to a pharmacologic AURKB inhibitor (Barasertib). Normalized growth rate measurements (such as GR$_{50}$) incorporate baseline cell division rate and are demonstrably more repro-ducible than the commonly calculated half-maximal inhibitor concentration (IC$_{50}$) when comparing different cell cultures (*Hafner et al., 2016*). An additional advantage of GR$_{50}$ is the ability to distin-guish growth rate inhibition (GR$_{50}$ approaches 0) versus cell death (GR$_{50}$ is negative). Since GR$_{50}$ calcu-lations require proliferation curves, we conducted live QPI over 48 hr with a range of Barasertib concentrations (0–300 nM). Primary melanocytes were largely resistant to the compound with reduced growth apparent only at supraphysiological levels (*Figure 5F*). In contrast, both melanoma cultures responded to tenfold lower dosage of the compound with GR$_{50}$ approaching zero, indicating primarily arrest, not death. In these conditions, the per cell dry mass also increased significantly, consistent with a G2/M arrest (*Figure 5G*). Taken together, these data demonstrate that AURKB inhibition limits the rate of BRAF$^{V600E}$-induced proliferation in human melanocytic nevi and melanomas.

## Discussion

Cellular senescence is conventionally defined as a permanent cell-cycle arrest associated with a variety of molecular markers and an irreversible arrest in the G0/G1 phase of the cell cycle (*He and Sharpless, 2017*; *Roy et al., 2020*; *Serrano et al., 1997*). The vast majority of melanocytic nevi are stably arrested and harbor BRAF$^{V600E}$, an oncogenic mutation that induces cellular senescence when expressed in a variety of mammalian cells (*Michaloglou et al., 2005*; *Serrano et al., 1997*; *Zhu et al., 1998*). While these observations seemingly implicate OIS as the mechanism driving nevus-associated proliferation arrest, a substantial body of evidence has challenged this paradigm in recent years (*Ruiz-Vega et al., 2020*; *King et al., 2009*; *Burian and Jemec, 2019*; *Cotter et al., 2007*; *Tran et al., 2012*; *Oaxaca et al., 2020*; *Zeng et al., 2018*; *Haferkamp et al., 2009*; *McNeal et al., 2015*). Here, we provide further evidence to challenge the model that acquisition of BRAF$^{V600E}$ induces premature cellular senescence in melanocytes and uncover an alternative mechanism driving nevus formation and stability. We propose that acquisition of the BRAF$^{V600E}$ mutation does not necessitate growth arrest, but instead permits melanocytes to toggle between hyperproliferation and mitotic arrest. Our transcriptomic analyses demonstrate that external stimuli can modulate the differentiation state of BRAF$^{V600E}$ melanocytes, orchestrate the expression level of MIR211-5p, and inform which phenotype is triggered by the oncogene—before or after the oncogene is introduced.

The senescence model is largely rooted in seminal work demonstrating that sustained ectopic expression of BRAF$^{V600E}$ induces proliferative arrest in primary human melanocytes (*Michaloglou et al., 2005*)—an experiment reproduced by several independent labs (*Haferkamp et al., 2009*; *McNeal et al., 2015*; *Leikam et al., 2008*; *Lu et al., 2016*), including our own (*Zeng et al., 2018*). In each of these studies, melanocytes were cultured in TPA-containing media. Thus, while observations presented here—that BRAF$^{V600E}$-induced proliferative arrest is reversible and conditional on the pres-ence of TPA— provide new insights that challenge the senescence model, they do not contradict the data presented in these previous reports.

One inherent limitation of using any in vitro approach to investigate the mechanisms underlaying clinical phenomena is an inherent incongruity in the magnitude of duration. Melanocytic nevi frequently persist for several decades, whereas in vitro experiments are generally conducted on the order of days to weeks. In previous in vitro studies, the periods of time melanocytes were monitored subsequent to BRAF$^{V600E}$ introduction ranged from 1 to 3 weeks. In studies utilizing an inducible RAF in human fibroblasts, 24 hr of expression was sufficient to drive a durable proliferative arrest, even upon removal of the oncogene (*Zhu et al., 1998*). In this study, we induced growth arrest with high levels of BRAF$^{V600E}$ for 30 days and observed the arrest was still reversible. It remains possible that an even longer exposure to BRAF$^{V600E}$ would elicit a permanent proliferative arrest consistent with senescence. It is noteworthy, however, that the identified mechanisms driving proliferative arrest in vitro were also supported by analyses of clinical nevi specimens.

While supported by analyses of both primary human melanocytes and clinical specimens, one limitation to this report is the lack of in vivo confirmation using animal models. One well-established genetically engineered mouse model of melanoma expresses BRAF$^{V600E}$ from the endogenous locus in tyrosinase-expressing cells with spatiotemporal control. The predominant phenotype of BRAF$^{V600E}$ activation in this model is melanocytic hyperplasia, which increases in prominence with BRAF$^{V600E}$ dose (*Dankort et al., 2009*). While growth-arrested melanocytic 'nevus-like' lesions also form in this model, they are not associated with markers of senescence (*Ruiz-Vega et al., 2020*). More recently, an elegant study utilizing a zebrafish model of melanoma demonstrated that driving BRAF$^{V600E}$ expression from promoters associated with melanocyte progenitors, but not differentiated melanocytes, resulted in melanoma initiation (*Arianna et al., 2021*). These observations are all consistent with our data and support the conclusion that BRAF$^{V600E}$-induced proliferation in melanocytes is dependent on differentiation state. Further studies will be required to confirm whether the microRNA mechanism reported here is conserved between human and model systems.

The presented model offers an explanation for several clinical phenomena. First, melanocytic nevi form when BRAF$^{V600E}$ drives hyperproliferation followed by proliferation arrest. Second, nevus eruption involves the expansion of previously stable nevi over a short period of time. Third, incomplete excision of melanocytic nevi can result in subsequent regrowth of the benign lesion (*King et al., 2009*). Each event requires interconversion of the BRAF$^{V600E}$ melanocyte between a proliferative and arrested phase. Our data demonstrate that BRAF$^{V600E}$ melanocytes can indeed oscillate between proliferative and arrested states and are consistent with each of these phenomena. Our interpretation assumes that once acquired, BRAF$^{V600E}$ expression is constant and that the environmental context of the melanocytes is variable, serving as a 'gatekeeper' to BRAF$^{V600E}$-induced proliferation. However, if BRAF$^{V600E}$ expression also fluctuates, then an equally plausible interpretation of our data is that BRAF$^{V600E}$ serves as a 'gatekeeper' to environmentally induced proliferation. Interestingly, nevus eruption has been observed in metastatic melanoma patients treated with single agent selective BRAF inhibitors (*Cohen et al., 2013*; *Chen et al., 2014*; *Chu et al., 2012*; *Dalle et al., 2013*; *Zimmer et al., 2012*), consistent with our observation that proliferation arrest in melanocytes requires sustained expression of the oncogene. A more thorough understanding of the responsible environmental stimuli in physiologic conditions, discussed in more detail below, would help to clarify which of the mechanisms—variable BRAF$^{V600E}$, variable environment, or both—underlie nevus formation, eruption, and recurrence.

It is important to note that while our data support a model of reversible proliferation arrest in nevi, we do not suggest this reversal is commonplace. Unquestionably, the vast majority of melanocytic nevi present with sustained arrest. However, we do propose that the duration of arrest is dynamic and modulated by exposure to external signals. Since our data suggest that loss of MIR211-5p and MIR328-3p expression and restoration of AURKB expression is concurrent with progression to melanoma, identification of the source of cell-extrinsic signals might suggest new strategies for chemoprevention or therapy. We have identified TPA, a known activator of PKC signaling, as one extrinsic factor that induces expression of MIR211-5p and a more dedifferentiated state. Previous studies have identified TPA as either a mitogen, inhibitor of the G1/S transition, or inhibitor of the G2/M transition, dependent on the context (*Coppock et al., 1995*; *Arita et al., 1998*; *Arita et al., 1992*; *Stavroulaki et al., 2008*; *Chao-Hsing and Hsin-Su, 1991*). We demonstrate that in vitro, TPA activation of MIR211-5p is required for BRAF$^{V600E}$-induced mitotic failure. Yet, the inclusion of TPA in primary melanocyte culture is artificial and the environmental stimulus

that regulates MIR211-5p expression in skin remains unknown. One hypothesis is that PKC activation from keratinocyte interactions serves as the extrinsic stimulus. MIR211-5p is reported to be transcriptionally regulated by both MITF (*Mazar et al., 2010*) and UV exposure (*Su et al., 2020*). However, TPA-free media containing an endothelin receptor type B ligand, endothelin-1, did not elicit MIR211-5p expression or BRAF$^{V600E}$ arrest, suggestive that the downstream signaling pathways regulated by the two molecules are not identical. PKC activation can also result in increased MAPK signaling (*Herrera et al., 1998*; *Ueda et al., 1996*) raising the possibility that internal MAPK fluctuations may play a role in toggling between BRAF$^{V600E}$-induced proliferation versus arrest. The term 'mitogenic window' describes the specific range of oncogene doses that elicit a hyperproliferative phenotype. The mitogenic window for MAPK signaling activation can be remarkably fine. Here, we conducted a twofold dilution series of doxycycline but observed no evidence of a mitogenic window. However, it remains possible that we simply missed the 'sweet spot' in TPA conditions, and that a finer dilution series would reveal a mitogenic window even in the presence of TPA. Regardless, our data demonstrate that in TPA conditions the mitogenic window for BRAF$^{V600E}$ expression is, at least, exceedingly narrow, whereas in non-TPA conditions the window is wide open. An important future direction will be to identify the nature and physiological source of this signal for melanocytic nevi in human skin.

Diminished MIR211-5p/MIR328-3p expression concurrent with increased AURKB expression occurs in a substantial portion of melanomas. AURKB expression was previously reported as significantly elevated in melanoma metastases (*Sharma et al., 2013*) and here we report increased expression in primary melanomas. Similarly, MIR211-5p expression is consistently observed as lost in melanoma as compared to melanocytic nevi across independent studies (*Babapoor et al., 2016*; *Torres et al., 2020*; *Latchana et al., 2016*). Indeed, MIR211-5p expression levels classify nevi from melanomas with a high degree of accuracy (*Babapoor et al., 2016*; *Torres et al., 2020*). We have reported on the diagnostic potential of MIR328-3p expression as well, although multi-study validation was impeded by the absence of hybridization probes in earlier microarray-based profiling data sets (*Torres et al., 2020*). In vitro, we have now demonstrated that inhibition of either miRNA is sufficient to block BRAF$^{V600E}$-induced arrest in melanocytes. These data suggest that mitotic failure serves as a barrier to melanoma progression and is consistent with the genome duplication and copy number alterations associated with invasive melanoma (*Shain et al., 2015*; *Vergara et al., 2021*). In melanoma, the specific cause of this genomic instability is unknown, but reentry of cells into cell cycle after mitotic failure has been shown to cause genome duplication and accumulation of copy number alterations in other cancers (*Telentschak et al., 2015*). Further experiments will be necessary to determine whether dysregulation of the spindle checkpoint in nevi contributes directly to the copy number variations present in melanomas.

In summary, we have identified that BRAF$^{V600E}$ induces a reversible arrest in human melanocytes orchestrated by MIR211-5p/MIR328-3p regulation of AURKB and conditional on the melanocyte differentiation state. We present a mechanistic model that allows for melanocytic nevus formation, eruption, and recurrence. Moreover, our observations provide an explanation for the genetic duplication and instability inherent to invasive melanomas and open new directions for novel chemopreventative and therapeutic strategies.

## Materials and methods

### Key resources table

| Reagent type (species) or resource | Designation | Source or reference | Identifiers | Additional information |
|---|---|---|---|---|
| Gene (*Homo sapiens*) | BRAF | https://www.ncbi.nlm.nih.gov/gene | Gene ID: 673 | |
| Cell line (*H. sapiens*) | Epidermal melanocytes (adult normal, adult nevus, and neonatal normal) | Other | Prep specific | Non-immortalized primary lines, derived from fresh skin. |
| Cell line (*H. sapiens*) | 501MEL, human melanoma | Boris Bastian | RRID:vCVCL_4633 | |
| Cell line (*H. sapiens*) | HCIMel019, human melanoma | This paper | HCIMel019 | Patient-derived melanoma line. Available from the HCI PRR core. |

*Continued on next page*

*Continued*

| Reagent type (species) or resource | Designation | Source or reference | Identifiers | Additional information |
|---|---|---|---|---|
| Biological sample (*H. sapiens*) | FFPE melanocytic tumor specimens | UCSF Dermatopathology Biospecimen Archive | De-identified | |
| Antibody | Anti-HSP90 (rabbit polyclonal) | CST | 4,874 | Western (1:1000) |
| Antibody | Anti-BRAFV600E (mouse monoclonal) | Spring Bioscience Corp | E19292; RRID:AB_11203851 | Western (1:1000) |
| Antibody | Anti-phosphoERK1/2 (rabbit monoclonal) | CST | 4,970 | Western (1:1000) |
| Antibody | Anti-AURKB (rabbit polyclonal) | Abcam | ab2254, RRID:AB_302923 | Western (1:1000); IHC (1:400) |
| Antibody | Anti-GPR3 (mouse monoclonal) | Abnova | H00002827-M01, RRID:AB_425462 | IHC/IF (1:125) |
| Antibody | Anti-AXL (rabbit monoclonal) | Cell Signalling Technology | 8661, RRID:AB_11217435 | IF (1:250) |
| Antibody | Anti-MLANA (mouse monoclonal) | Abcam | ab731, RRID:AB_305836 | IF (1:100) |
| Recombinant DNA reagent | pTRIPZ-diBRAFV600E | Todd Ridky | | gift |
| recombinant DNA reagent | pLVX-AURKB | This paper | Addgene #153,316 | See Materials and Methods: Generation of Lentiviral vectors. Available from Addgene. |
| recombinant DNA reagent | pLVX-GPR3 | This paper | Addgene #153,317 | See Materials and Methods: Generation of Lentiviral vectors. Available from Addgene. |
| recombinant DNA reagent | pLVX-anti-MIR211-5p | This paper | Addgene #153,318 | See Materials and Methods: Generation of Lentiviral vectors. Available from Addgene. |
| recombinant DNA reagent | pLVX-anti-MIR328-3p | This paper | Addgene #153,319 | See Materials and Methods: Generation of Lentiviral vectors. Available from Addgene. |
| recombinant DNA reagent | pLVX-Che-zsGreen | This paper | Addgene #153,320 | See Materials and Methods: Generation of Lentiviral vectors. Available from Addgene. |
| sequence-based reagent | hsa-MIR211-5p | Dharmacon | C-300566-03-0005 | |
| sequence-based reagent | hsa-MIR328-3p | Dharmacon | C-300695-03-0005 | |
| sequence-based reagent | MIRControl 1 (Control RNA) | Dharmacon | CN-001000-01-05 | |
| commercial assay or kit | ALDEFLUOR kit | Stemcell Technologies | 01700 | |
| chemical compound, drug | barasertib | Selleckchem | A1147 | |
| chemical compound, drug | doxycycline | Sigma | D9891 | |

## Transcriptomic profiling

For FFPE samples, histopathologic review, microdissection, targeted exon sequencing, phylogenetic analysis, and RNA and small RNA sequencing of each tumor area were previously described (phs001550.v2.p1) (*Zeng et al., 2018*; *Shain et al., 2018*; *Torres et al., 2020*). For this study, small RNA reads were aligned to human reference (hg37) with Bowtie (*Langmead et al., 2009*) and small RNA reference groups (miRBase21) were counted. For mRNA sequencing, reads were aligned to human reference (hg37) with Bowtie2 and counted with HTSeq. DE analysis was performed using DESeq2 (*Love et al., 2014*) with p values adjusted by the Benjamini-Hochberg method (p-adj). Previously published RNA sequencing data sets from *Shain et al., 2018* and *Torres et al., 2020* were re-analyzed here with DESeq2 and combined for visualization. For RNA profiling from primary melanocytes, total RNA was extracted using TRIzol Reagent (Thermo Fisher Scientific, 15596-026) and further purified with the RNeasy Power Clean Pro Cleanup Kit (Qiagen, 13997-50) to remove

melanin. For small RNAseq of nevus melanocytes sequencing libraries were constructed with the TailorMix Small RNA Library Preparation Kit (SeqMatic, CA) and sequencing was performed on the Illumina HiSeq2500 platform at single-end 50 bp. For small RNAseq of cultured BRAF$^{V600E}$ transduced melanocytes, sequencing libraries were constructed with the Qiagen QIAseq miRNA Library Prep Kit, and sequencing was performed on the NovaSeq SP platform at paired-end 50 bp. For mRNAseq of microRNA mimic nucleofected melanocytes, 150 bp paired-end sequencing was conducted by GeneWiz. For mRNAseq of cultured BRAF$^{V600E}$ transduced melanocytes, sequencing libraries were constructed with the Illumina TruSeq Stranded mRNA Library Prep Kit, and sequencing was performed on the NovaSeq S4 platform at paired-end 150 bp.

## Cell derivation and culture

Benign human melanocytic nevi were excised with informed consent from patient donors at the UCSF Dermatology clinic (San Francisco, CA) or HCI Dermatology clinic (Salt Lake City, UT) according to Institutional Review Board approved protocols. Nevus tissue was minced and enzymatically dissociated in 1 mg/ml collagenase (Sigma-Aldrich, 11213857001) and 3.3 mg/ml dispase (Sigma-Aldrich, 4942078001) in DMEM (Thermo Fisher Scientific, 10569044) for 1 hr at 37°C. Melanocytes were further isolated by 5 days exposure to 10 μg/ml G418 (InvivoGen, ant-gn-1). BRAF status was confirmed via sanger sequencing (Quintarabio) using the primer set: BRAF forward: 5'-GCA CGA CAG ACT GCA CAG GG -3'; BRAF reverse: 5'-AGC GGG CCA GCA GCT CAA TAG-3'. BRAF wild-type (normal) human melanocytes were isolated from de-identified and IRB consented neonatal foreskins or adult skin. Foreskin tissue was incubated overnight at 4°C in dispase and epithelia were mechanically separated from the dermis the following morning. Epithelial tissue was minced and incubated in 0.25% trypsin (Gibco 25200056) for 4 min at 37°C. Trypsin was quenched and tissue centrifuged at 500×$g$ for 5 min at room temperature. The cell/tissue pellet was resuspended in Melanocyte medium (Thermo Fisher Scientific, M254500) with HMGS (Thermo Fisher Scientific, S0025) and plated in low volume. Melanocytes were expanded in Melanocyte medium with HMGS and assayed in either Melanocyte medium with HMGS (+TPA) or Melanocyte medium with HMGS-2 (TPA-free, Thermo Fisher Scientific, S0165) as indicated.

The 501Mel human melanoma line (Gift from Dr. Boris Bastian, CVCL_4633, authenticated via STR profiling) were cultured in RPMI with 10% fetal bovine serum (FBS; VWR, S107G), 1% penicillin-streptomycin (Gibco, 15140122), 1% L-glutamine (Gibco, 25030149), and 1% non-essential amino acids (Gibco, 11140050). HCIMel019 was derived from patient-derived xenograft tumors propagated in mice. An HCIMel019 (P2) subcutaneous tumor was resected from mouse, minced in digestion buffer (100 μM HEPES (Gibco, 15630-080), 5% FBS (DENVILLE, FB5001-H), 20 μg/ml gentamicin (Gibco, 15710-064), 1× insulin (Gibco, 51500-056), and 1 mg/ml collagenase IV (Gibco, 17104-019) in DMEM (Gibco, 11965-092)), and digested overnight at 37°C. Cells were filtered through a 100 μm filter (Falcon, 352360) and red blood cells were removed with RBC lysis buffer (0.5 M EDTA, 0.5 M KHCO3 [Sigma-Aldrich, 237205], and 5 M NH4CL [Sigma-Aldrich, A9434]). Remaining cells were washed with phosphate-buffered saline (PBS; Gibco, 10010023), and cultured at 37°C and 5% CO2 in Mel2 media consisting of 80% MCDB153 (Sigma-Aldrich, M7403), 20% L15 (Gibco, 11415-064), 2% FBS (DENVILLE, FB5001-H), 1.68 mM CaCL (Sigma-Aldrich, C4901), 1× insulin (Gibco, 51500-056) 5 ng/ml EGF (Sigma-Aldrich, E9644), 15 μg/ml Bovine Pituitary Extract (Gibco 13028-014), and 1× Pen/strep (Gibco, 15070-063). Cell cultures were tested monthly for mycoplasma contamination (ATCC).

## Small RNA nucleofection

Human melanocytes were trypsinized (Gibco, 25300062), quenched, and centrifuged at 300×$g$. Cells were resuspended in R Buffer (Thermo Fisher Scientific, MPK1025) at 10,000 cells/μL. About 10 μl of cell slurry was mixed with miRIDIAN microRNA mimics (Dharmacon: hsa-MIR211-5p C-300566-03-0005, hsa-MIR328-3p C-300695-03-0005, MIRControl 1 CN-001000-01-05, or MIRControl 2 CN-002000-01-05); or On-TargetPlus siRNA (Dharmacon, custom library); at 4 μM final concentration and nucleofected using the NEON Transfection System and protocol (Thermo Fisher Scientific, MPK5000).

## Generation of Lentiviral vectors

pTRIPZ-diBRAFV600E was a gift from Todd Ridky (*McNeal et al., 2015*). pLVX-AURKB (Addgene #153316) and pLVX-GPR3 (Addgene #153317) were generated by subcloning the respective human cDNA (from Addgene #100142 and #66350) into the MluI and BamHI sites of the pLVX-Che-hi3 vector (a gift of Sanford Simon) (*Takacs et al., 2017*). pLVX-anti-MIR211-5p (Addgene #153318), pLVX-anti-MIR328-3p (Addgene #153319), and pLVX-Che-zsGreen (Addgene #153320) were generated by inserting zsGreen with or without a 3'UTR into the MluI and XbaI sites of the pLVX-Che-hi3 vector. The 3'UTRs contained seven tandem microRNA binding sites to report and inhibit microRNA function, as previously described (*Judson et al., 2013*). ZsGreen was subcloned from pHIV-zsGreen gift from Bryan Welm & Zena Werb (Addgene #18121) (*Welm et al., 2008*).

## Lentiviral transduction

$2.75 \times 10^6$ HEK293T cells were plated on 10 cm tissue culture dishes and grown for ~24 hr in DMEM with 10% FBS. For each 10 cm plate, 5 µg lentiviral vector, 3.3 µg of pMLV-GagPol, and 1.7 µg of pVSV-G packaging plasmids were added to 500 µl of jetPRIME buffer and 20 µl of jetPRIME transfection reagent (Polyplus, 712-60), and transfected according to the manufacturer's instructions. 48 hr post-transfection, viral supernatant was collected and filtered using 0.45 µm syringe filters (Argos 4395-91). Human melanocytes seeded at $1.0 \times 10^5$–$2.0 \times 10^5$ cells/well density in six-well plates were incubated in viral supernatant with 10 µg/ml polybrene (Sigma-Aldrich, TR-1003) and centrifuged at $300 \times g$ for 60 min at room temperature. The viral supernatant was removed and replaced with growth media. Transduced cells were either selected with puromycin (1 µg/mL for 5 days, Sigma-Aldrich P8833) or sorted for mCherry expression using a BD FACSAria II. Cells expressing pTRIPZ-diBRAFV600E were treated with doxycycline (Sigma-Aldrich, D9891) at indicated concentrations.

## Live quantitative phase imaging

Live quantitative imaging was performed using either the HoloMonitor M4 imaging cytometer (Phase Holographic Imaging, Lund, Sweden) or the Livecyte platform (Phasefocus, Sheffield, UK). Analyses of cell proliferation, dry cell mass, and death were acquired with the M4 platform and analyzed using HStudio (v2.6.3) as previously described (*Hejna et al., 2017*). For each experiment, human melanocytes were seeded into six-well plates (Sarstedt, 83.3920) at 100,000 cells/well and either live-imaged or serially imaged as indicated. Analysis of growth rate was acquired with the M4 platform and analyzed using App Suite (v3.2.0.60) as previously described (*Hafner et al., 2016*). For each experiment, 100,000 melanocytes, 60,000 501Mel cells, or 150,000 HCIMel019 cells were plated per well and media containing indicated concentrations of barasertib (AURKB inhibitor; AZD1152-HQPA | AZD2811, Selleckchem, A1147) was added. Cells were imaged for 48–72 hr. Analyses of fluorescent reporters coupled with QPI were conducted using the Livecyte platform and were analyzed using Analyze (v3.1) (Phasefocus, Sheffield, UK).

## EdU assays

Normal human melanocytes were nucleofected with microRNA mimicsor infected with lentiviral vectors (as above) and seeded in 48-well plates at a density of 50,000 cells/well. Four days post-seeding, EdU was added to culture media at a 10 µM final concentration. 24 hr after EdU addition, media was removed and cells were stained for EdU incorporation and nuclei using the Click-iT EdU Imaging Kit (Thermo Fisher Scientific, C10337) and protocol. Images of EdU and nuclei staining were acquired using the Evos FL microscope and quantified with Fiji.

## Western blotting

Protein was collected using RIPA Buffer (Thermo Fisher Scientific, 89901) supplemented with HALT Protease and Phosphatase Inhibitor (Thermo Fisher Scientific, 78446). Immunoblotting was carried out as previously described. Membranes were incubated overnight at 4°C with primary antibodies at the following dilutions: anti-HSP90 (CST, 4874) 1:1000, anti-BRAFV600E (Spring Bioscience Corp, E19292) 1:1000, anti-phosphoERK1/2 (CST, 4970) 1:1000, and anti-AURKB (Abcam, ab2254) 1:1000. Membranes were washed 4× with TBS and 0.5% Tween20, incubated with HRP-conjugated secondary

antibody (1:2000) for 30 min at room temperature, and visualized with Lumina Forte Western HRP substrate (Millipore, WBLUF0500).

### Flow analyses

For assessing DNA content, trypsinized melanocytes (as above) were resuspended in 400 PBS. 1 ml – 20°C 200 proof ethanol was added dropwise while gently vortexing to achieve 70% final concentration for fixation and incubated overnight at – 30°C. Cells were centrifuged at 800×g for 5 min, washed with fresh PBS, and incubated in 500 µL FxCycle PI/RNase Staining Solution (Invitrogen, F10797) for 20 min at 37°C. Data were collected on the Fortessa (BD) at low speed and analyzed with FlowJo v10.7.1. Annexin V/PI staining was performed by trypsinizing melanocytes as above. Cells were washed with PBS and resuspended with Annexin V binding buffer (BioLegend, 422201) at a concentration of $1 \times 10^6$ cell/ml. 5 µl APC Annexin V antibody (BioLegend, 640919) and 10 µl PI were added. Cells were gently vortexed and incubated in the dark at room temperature for 15 min. Cells were resuspended in 400 µl of Annexin V Binding Buffer. Data were collected on the FACs Verse (BD) and were analyzed with FlowJo v10.7.1. Aldehyde dehydrogenase activity was measured first by trypsinizing melanocytes as above and then using the ALDEFLUOR Kit (Stemcell Technologies, 01700) as described in the manufacture's protocol.

### Human nevus and melanoma tissue immunohistochemistry

The archived clinical specimens used in the study were procured with IRB approval from the UCSF Dermatopathology Pathology Service archives. For assessment of AURKB and GPR3 expression, 11 melanoma and 11 melanocytic nevi with previous diagnosis were de-identified and re-verified histologically (UEL). For assessment of DNA content, 15 additional melanocytic nevi with previous diagnosis were re-verified histologically (UEL). Tissue was fixed in 10% neutral-buffered formalin, processed, embedded in paraffin, and stained with hematoxylin and eosin. 4 µm FFPE sections were stained with AURKB (1:400 dilution, Abcam ab2254) or GPR3 (1:125 dilution, Abnova H00002827-M01) antibodies. UEL reviewed immunohistochemical stains with semiquantitative grading for GPR3 (0=none; 1=patchy positive; 2=strong positive) and AURKB (0=none; 1=rare positive; 2=scattered positive; 3 = frequent positive; 4=many positive). For in vitro immunofluorescence, cells were fixed and stained as previously described with AXL (1:250 dilution, Cell Signaling Technology #8661) or MLANA (1:100, Abcam ab731).

### Statistical analyses

For differential gene expression, p values were calculated with the DESeq2 (v1.30.1) default Wald test adjusted by the Benjamini-Hochberg method using a 5% false discovery rate (FDR) (**Benjamini et al., 2001**). Pathway analyses were analyzed using the fast gene set enrichment package (**Korotkevich et al., 2016**) in R with a 10% FDR. Gene set enrichment analyses were conducted using GSEA (v4.0.3, Broad Institute) Preranked tool with 1000 permutations. For in vitro experiments, pilot studies were initially conducted in triplicate. The required minimal sample size to assess the difference between independent means with independent standard deviations with an alpha error probability of 0.05 and a power of 0.95 was calculated using G*Power (v. 3.1.9.4). P values were calculated using either paired or unpaired two-tailed t-tests or Wilcoxon tests via Prism 8 (GraphPad) as indicated in figure legends.

### Experimental set-up

For cohorts of clinical specimens, sample number (n) refers to the number of specimens, each from a different patient. For in vitro experiments, n refers to independent experiments conducted on different days. In cases where fresh primary melanocytes were used, the cells in each experiment are derived from an independent donor. In cases where cell lines are used, the cells in each experiment represent a different passage and/or day of that line. For different conditions within experiments, replicate wells were plated and arbitrarily chosen for each condition (e.g., not treated versus treated; different concentrations of a compounds, etc.). All data were included.

## Acknowledgements

The authors acknowledge the use of the HCI Shared Resources for Research Informatics (RI), Cancer Biostatistics (CB), High-Throughput Genomics and Bioinformatics Analysis (GBA), and the Biorepository and Molecular Pathology (BMP) Research Histology Section supported by P30CA042014 awarded to HCI from the National Cancer Institute.

## Additional information

### Competing interests

Maria L Wei, Robert L Judson-Torres: RLJ and MLW are inventors on international patent application no. PCT/US2019/023834 concerning the use of microRNAs investigated in this article as molecular diagnostics for melanoma. The other authors declare that no competing interests exist.

### Funding

| Funder | Grant reference number | Author |
|---|---|---|
| National Institutes of Health | DP5OD019787 | Robert L Judson-Torres |
| National Cancer Institute | R01CA229896 | Robert L Judson-Torres |
| National Cancer Institute | 5F31CA236377 | Andrew S McNeal |
| National Cancer Institute | P30CA042014 | Robert L Judson-Torres |
| Program for Breakthrough Biomedical Research | Sandler Fellowship | Robert L Judson-Torres |
| 5 for the Fight | 5 for the Fight Fellowship | Robert L Judson-Torres |

The funders had no role in study design, data collection and interpretation, or the decision to submit the work for publication.

### Author contributions

Andrew S McNeal, Formal analysis, Funding acquisition, Investigation, Visualization, Writing – original draft; Rachel L Belote, Investigation, Methodology, Resources, Writing – review and editing; Hanlin Zeng, Investigation, Methodology, Visualization, Writing – review and editing; Marcus Urquijo, Kendra Barker, Investigation; Rodrigo Torres, Meghan Curtin, Formal analysis; A Hunter Shain, Data curation, Writing – review and editing; Robert HI Andtbacka, Douglas Grossman, Maria L Wei, Resources, Writing – review and editing; Sheri Holmen, Matt W VanBrocklin, Conceptualization; David H Lum, Conceptualization, Writing – review and editing; Timothy H McCalmont, Resources; Ursula E Lang, Formal analysis, Investigation, Resources, Writing – review and editing; Robert L Judson-Torres, Conceptualization, Formal analysis, Funding acquisition, Investigation, Supervision, Visualization, Writing – original draft

### Author ORCIDs

Robert L Judson-Torres http://orcid.org/0000-0002-6559-0553

### Ethics

Human subjects: The archived clinical specimens used in the study were procured with IRB approval from the UCSF Dermatopathology Pathology Service archives (11-07951). Fresh human tissue samples used in this study were obtained with informed consent and consent to publish from each patient. Samples were first coded before being provided to the researchers. The study protocol conformed to the ethical guidelines of the Declaration of Helsinki (1975). Samples were processed following standard operating procedures approved by IRB at University of California, San Francisco (16-20299) and at the Huntsman Cancer Institute (00124195).

### Decision letter and Author response

Decision letter https://doi.org/10.7554/eLife.70385.sa1
Author response https://doi.org/10.7554/eLife.70385.sa2

## Additional files

### Supplementary files
• Supplementary file 1. Results of differential expression analyses performed on previously published databases of matched melanocytic nevi and melanoma-arising-from-nevi.

• Supplementary file 2. Normalized read counts from small RNA sequencing performed on melanocytes derived from healthy human skin and melanocytic nevi.

• Supplementary file 3. Results of differential expression analyses performed on primary melanocytes nucleofected with microRNA or control mimics.

• Supplementary file 4. Results from siRNA screen for EdU incorporation performed in primary melanocytes. Mean and standard deviation of triplicate screens shown for each gene. Column D indicates the microRNA predicted to targeted the gene.

• Supplementary file 5. Results of differential expression analyses performed on primary melanocytes with or without BRAF$^{V600E}$ in+ TPA or TPA-free conditions.

• Supplementary file 6. Results of GSEA analyses performed on primary melanocytes with or without BRAF$^{V600E}$ in+ TPA or TPA-free conditions.

• Transparent reporting form

### Data availability
The sequencing data produced in this manuscript is available from GEO under the accession number GSE150849. Source data for other large datasets are provided as supplementary files. Other data are included in the manuscript and figures.

The following dataset was generated:

| Author(s) | Year | Dataset title | Dataset URL | Database and Identifier |
|---|---|---|---|---|
| Judson-Torres R, McNeal A | 2020 | microRNAs Restrain Proliferation in BRAFV600E Melanocytic Nevi | https://www.ncbi. nlm.nih.gov/geo/ query/acc.cgi?acc= GSE150849 | NCBI Gene Expression Omnibus, GSE150849 |

The following previously published datasets were used:

| Author(s) | Year | Dataset title | Dataset URL | Database and Identifier |
|---|---|---|---|---|
| Bastian BC, Shain AH, Judson RL | 2015 | The genetic and transcriptomic evolution of melanoma | https://www.ncbi.nlm. nih.gov/projects/gap/ cgi-bin/study.cgi? study_id=phs001550. v2.p1 | NCBI Gene Expression Omnibus, phs001550.v2.p1 |

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
