## [Decision Letter]

**Acceptance summary:**

The authors use primary human melanocytes and clinical samples to show that microRNA-dependent suppression of the mitotic kinase AURKB determines whether mutations activating the BRAF oncogene initiate proliferative tumor growth, or is growth suppressive through induction of senescence. In addition, the levels of the microRNAS are themselves dependent on microenvironmental controls. This manuscript would appeal to researchers in the melanoma field, especially those studying the underlying mechanisms behind phenotypic plasticity and tumor heterogeneity.

**Decision letter after peer review:**

Thank you for submitting your article "BRAF V600E induces reversible mitotic arrest in human melanocytes via microRNA-mediated suppression of AURKB" for consideration by *eLife*. Your article has been reviewed by 2 peer reviewers, and the evaluation has been overseen by a Reviewing Editor and Erica Golemis as the Senior Editor. The reviewers have opted to remain anonymous.

Essential revisions:

1. The section dealing with the influence of TPA on melanocytes is especially interesting, but not always easy to follow. Some issues related to clarity should be addressed. Briefly, TPA and BrafV600E represent two independent inputs into melanocytes, and the behaviors that result from activating them separately or together are clearly non-additive. The most unbiased way to look at the data is from the standpoint of a 2x2 matrix of four conditions, with TPA+ or – on one axis and BRAFV600E + or – on the other. The authors present some data in this format, while in other cases they hold one condition constant and vary the other. From the standpoint of presentation, this helps the authors build their case that TPA essentially "gates" the response to Braf. Of course, one could just as easily use the same data, to support the case that Braf "gates" the response to TPA. Neither one is more or less true than the other, and given that TPA is presumed to be a stand-in for some as-yet-unknown environmental factor that melanocytes encounter in vivo, it is difficult to say which perspective is the more useful. Although I would not go so far as to suggest that the authors change the order in which data are discussed – it flows very logically as it is – it would be helpful to see some sort of summary figure that re-frames all of the observations into the simple 2x2 matrix format, so that readers could consider their own interpretations.

2a. Throughout the work, it is noteworthy that most proliferation assays are rather short-term. If there's a reason for that (i.e. affects on proliferation are only transient) that should be discussed. In particular, the following questions related to behaviors over time should be resolved:

2b. Although recent studies have called into question the view that nevi arrest by senescence, studies done in vitro based on the introduction of BrafV600E into melanocytes have suggested that, at least in vitro, they are driven into senescence. Are the authors disputing the results of those studies, or are there methodological differences to explain the discrepancy (e.g. other groups cultured cells for longer or in different media)?

2c. One implication of the data in Figure 3 is that, if not for the common use of TPA in culture medium, introduction of BrafV600E into melanocytes would result in durable (long-lasting) stimulation of proliferation. Although this paper addresses the early responses of melanocytes, it would be valuable to state whether those responses persist or not.

3. Several independent results in the paper support the view that the proliferation of non-BRAFV600E-expressing melanocytes doesn't depend at all on AURKB function, which is surprising given the key role attributed to Aurkb in mitosis. Do the authors think there is an alternate pathway of cell cycle control operating in these cells? Or do they think it's more of a quantitative matter, e.g. that modest, low-level proliferation can be sustained without (much) AURKB but a more rapid, aggressive proliferative state requires it?

4. The limited sample sizes (n = 3) in earlier experiments in addition to the lack of in vivo work lessens the potential clinical impact this manuscript could have on the melanoma community. If it is possible to extend the sample size for some of this work, it would improve the importance of the study.

5. Some very specific questions that the authors should address are the following:

Figure 2E: the detected miRNA binding sites in AURKB and GPR3 seem a little weak; how do these sequences rank among other detected targets for these miRNAs in the genome?

Figure 3G: Notwithstanding the presentation value of displaying cell number relative to the dox-negative control, doing so suppresses information about how TPA itself affects cell number. Thus, the control levels to which data were normalized should be given, at least in the figure legend.

Figure 4C Data availability: Although the PCA data are shown for four separate conditions: plus-or-minus TPA and plus-or-minus Dox, the underlying mRNA data in Supplemental Table 5 merge pairs of conditions (No Dox vs. DoxAll, and TPAAll vs TPAFreeAll). The authors do present the microRNA data separated into four conditions. They should do the same with the mRNA data.

Figure 5F: The published IC50 for Barasertib for Aurkb is 0.36 nM, but the half-effective dose in Figure 5F seems to be about 12-25 nM. Do the authors have an explanation for why the difference is so large?

*Reviewer #1 (Recommendations for the authors):*

Addressing the following comments and suggestions would improve the presentation of the paper:

1. The section dealing with the influence of TPA on melanocytes is especially interesting, but not always easy to follow. Some issues related to clarity should be addressed. Briefly, TPA and BrafV600E represent two independent inputs into melanocytes, and the behaviors that result from activating them separately or together are clearly non-additive. The most unbiased way to look at the data is from the standpoint of a 2x2 matrix of four conditions, with TPA+ or – on one axis and BRAFV600E + or – on the other. The authors present some data in this format, while in other cases they hold just one condition constant and vary the other. From the standpoint of presentation, this helps the authors build their case that TPA essentially "gates" the response to Braf. Of course, one could just as easily use the same data, to support the case that Braf "gates" the response to TPA. Neither one is more or less true than the other, and given that TPA is presumed to be a stand-in for some as-yet-unknown environmental factor that melanocytes encounter in vivo, it is difficult to say which perspective is the more useful. Although I would not go so far as to suggest that the authors change the order in which data are discussed-it flows very logically as it is-it would be nice to see some sort of summary figure that re-frames all of the observations into the simple 2x2 matrix format, so that readers could consider their own interpretations.

2a. Throughout the work, it is noteworthy that most proliferation assays are rather short-term. If there's a reason for that (i.e. proliferation is only transient) that should be discussed. In particular, the following questions related to behaviors over time should be resolved:

2b. Although recent studies have called into question the view that nevi arrest by senescence, studies done in vitro based on the introduction of BrafV600E into melanocytes have suggested that, at least in vitro, they are driven into senescence. Are the authors disputing the results of those studies, or are there methodological differences to explain the discrepancy (e.g. other groups cultured cells for longer or in different media)?

2c. One implication of the data in Figure 3 is that, if not for the common use of TPA in culture medium, introduction of BrafV600E into melanocytes would result in durable (long-lasting) stimulation of proliferation. Although this paper addresses the early responses of melanocytes, it would be valuable state whether those response persist or not.

3. Several independent results in the paper support the view that the proliferation of non-BRAFV600E-expressing melanocytes doesn't depend at all on AURKB function, which is surprising given the key role attributed to Aurkb in mitosis. Do the authors think there is an alternate pathway of cell cycle control operating in these cells? Or do they think it's more of a quantitative matter, e.g. that modest, low-level proliferation can be sustained without (much) AURKB but a more rapid, aggressive proliferative state requires it?

4. Some more specific questions that the authors should address are the following:

Figure 2E: the detected miRNA binding sites in AURKB and GPR3 seem a little weak; how do these sequences rank among other detected targets for these miRNAs in the genome?

Figure 3G: Notwithstanding the presentation value of displaying cell number relative to the dox-negative control, doing so suppresses information about how TPA itself affects cell number. Thus, the control levels to which data were normalized should be given, at least in the figure legend.

Figure 4C Data availability: Although the PCA data are shown for four separate conditions: plus-or-minus TPA and plus-or-minus Dox, the underlying mRNA data in Supplemental Table 5 merge pairs of conditions (No Dox vs. DoxAll, and TPAAll vs TPAFreeAll). The authors do present the microRNA data separated into four conditions. They should do the same with the mRNA data.

Figure 5F: The published IC50 for Barasertib for Aurkb is 0.36 nM, but the half-effective dose in Figure 5F seems to be about 12-25 nM. Do the authors have an explanation for why the difference is so large?

---

## [Author Response]

Essential revisions:1. The section dealing with the influence of TPA on melanocytes is especially interesting, but not always easy to follow. Some issues related to clarity should be addressed. Briefly, TPA and BrafV600E represent two independent inputs into melanocytes, and the behaviors that result from activating them separately or together are clearly non-additive. The most unbiased way to look at the data is from the standpoint of a 2x2 matrix of four conditions, with TPA+ or – on one axis and BRAFV600E + or – on the other. The authors present some data in this format, while in other cases they hold one condition constant and vary the other. From the standpoint of presentation, this helps the authors build their case that TPA essentially "gates" the response to Braf. Of course, one could just as easily use the same data, to support the case that Braf "gates" the response to TPA. Neither one is more or less true than the other, and given that TPA is presumed to be a stand-in for some as-yet-unknown environmental factor that melanocytes encounter in vivo, it is difficult to say which perspective is the more useful. Although I would not go so far as to suggest that the authors change the order in which data are discussed – it flows very logically as it is – it would be helpful to see some sort of summary figure that re-frames all of the observations into the simple 2x2 matrix format, so that readers could consider their own interpretations.

We thank the reviewers for the insightful point. We had based our previous interpretations on the assumption that in the clinical presentation of nevi and melanomas, even if not in our in vitro system, once the BRAF mutation was acquired it was constant, but the environment might change. But you are absolutely correct, if expression of the oncogene can vary (and why not?), then either interpretation is equally as valid. From our perspective, either interpretation is also equally as interesting. We personally favor the interpretation that the transcriptional landscape “gatekeeps” the growth-arrest phenotype because the mechanism we’ve identified here – MIR211-5p expression – is downstream of the media conditions, not mutant BRAF expression – but it’s the observation that the phenotype is reversible and conditional at all we find most interesting. We also recognize the importance of providing both perspectives for the reader to consider. To better acknowledge both interpretations:

First, we discuss both interpretations in the Discussion (lines 758-767):

“Our interpretation assumes that once acquired, BRAF^V600E^ expression is constant and that the environmental context of the melanocytes is variable, serving as a “gatekeeper” to BRAF^V600E^-induced proliferation. However, if BRAF^V600E^ expression also fluctuates, then an equally plausible interpretation of our data is that BRAF^V600E^ serves as a “gatekeeper” to environmentally induced proliferation. Interestingly, nevus eruption has been observed in metastatic melanoma patients treated with single agent selective BRAF inhibitors^62–66^, consistent with our observation that proliferation arrest in melanocytes requires sustained expression of the oncogene. A more thorough understanding of the responsible environmental stimuli in physiologic conditions, discussed in more detail below, would help to clarify which of the mechanisms – variable BRAF^V600E^, variable environment, or both – underlie nevus formation, eruption and recurrence.”

Second, we have provided an updated Supplementary File 5 that includes all differential expression for pairwise comparisons of all conditions and a new Supplementary File 6, which includes GSEA enrichment results for all of the pairwise comparisons, for the reader to consider.

Third, we have attempted to present more of the data, as suggested, as 2x2 matrics. In the original submission, the GSEA analysis results were not provided as a matrix – this was an intentional choice as the GSEA approaches we are familiar with require a pair-wise comparison. We originally ran all sets of pair-wise GSEAs and noticed that almost all significant enrichments were true in either both TPA conditions or both dox conditions. Therefore, for the sake of clearly presenting the data, we presented the data by holding one condition constant. However, there are a few cases of where the two stimuli converged – which we now discuss in lines 420-455. The new Figure 4—figure supplement 1 is our attempt at summarizing the set of pair-wise comparisons to 2x2 matrices. To further ensure the reader can make their own interpretations, Supplementary File 6, as mentioned above, contains all enrichments for all comparisons.

Finally, we updated our summary panel (Figure 4J) to indicate that change in BRAF expression can also change the phenotype (labels and double headed arrows). This panel, itself a matrix, combined with the afore mentioned Figure 4 —figure supplement 1, summarizes our observations is the 2x2 format requested.

We believe these changes provide the data required for the reader to generate their own interpretations while also acknowledging both interpretations as a possibility.

2a. Throughout the work, it is noteworthy that most proliferation assays are rather short-term. If there's a reason for that (i.e. affects on proliferation are only transient) that should be discussed. In particular, the following questions related to behaviors over time should be resolved:

When designing the experiments, we chose durations of BRAF induction that are consistent with most previous studies exploring this topic. To further push the system, we have now included additional experiments presented in Figure 3I-K, whereby we treated the melanocytes with dox for 30 days, prior to either removing the dox or removing the TPA. The melanocytes remained arrested for the 30-day duration prior to proliferating again once Dox or TPA was removed.

With this additional experiment, our durations are now equal to or greater than that usually published by the field. However, we also appreciate the obvious discrepancies between any reasonable in vitro experiment and a melanocytic nevus that may persist for many decades. Longer experiments could yield different results, but we would be hard pressed to justify how long is “long enough” to truly model decades. It is for this reason that we were excited to find evidence for the mechanism discovered in vitro in the clinical specimens we analyzed.

To ensure readers appreciate all of these points, we have now included the following paragraphs in the discussion (lines 719-729):

“One inherent limitation of using any in vitro approach to investigate the mechanisms underlaying clinical phenomena is an inherent incongruity in the magnitude of duration. Melanocytic nevi frequently persist for several decades, whereas in vitro experiments are generally conducted on the order of days to weeks. In previous in vitro studies, the periods of time melanocytes were monitored subsequent to BRAF^V600E^ introduction ranged from one to three weeks. In studies utilizing an inducible RAF in human fibroblasts, 24 hours of expression was sufficient to drive a durable proliferative arrest, even upon removal of the oncogene^17^. In this study, we induced growth arrest with high levels of BRAF^V600E^ for thirty days and observed the arrest was still reversible. It remains possible that an even longer exposure to BRAF^V600E^ would elicit a permanent proliferative arrest consistent with senescence. It is noteworthy, however, that the identified mechanisms driving proliferative arrest in vitro were also supported by analyses of clinical nevi specimens.”

2b. Although recent studies have called into question the view that nevi arrest by senescence, studies done in vitro based on the introduction of BrafV600E into melanocytes have suggested that, at least in vitro, they are driven into senescence. Are the authors disputing the results of those studies, or are there methodological differences to explain the discrepancy (e.g. other groups cultured cells for longer or in different media)?

Our data are consistent with previous in vitro studies when considering the data presented in those studies, albeit not always when considering the authors’ interpretations of those data. The previous studies we are aware of (our papers included) all include TPA in their media, and thus our observation that BRAFV600E + TPA = growth arrest is entirely consistent. Removal of TPA from the media is the most notable methodological difference. Other possibilities included the duration of the experiment or the dose of BRAF expression – both of which we address in new Figures3I-K. In short, we do not believe there are discrepancies between our observations and the previous studies, and we have added to our understanding of the system by exploring altering variables that were not previously considered.

To make these points clear, we have added the following discussion to the manuscript (lines 712-717):

“The senescence model is largely rooted in seminal work demonstrating that sustained ectopic expression of BRAF^V600E^ induces proliferative arrest in primary human melanocytes^9^ – an experiment reproduced by several independent labs^24,25,59,60^, including our own^23^. In each of these studies, melanocytes were cultured in TPA containing media. Thus, while observations presented here – that BRAF^V600E^ -induced proliferative arrest is reversible and conditional on the presence of TPA – provide new insights that challenge the senescence model, they do not contradict the data presented in these previous reports.”

2c. One implication of the data in Figure 3 is that, if not for the common use of TPA in culture medium, introduction of BrafV600E into melanocytes would result in durable (long-lasting) stimulation of proliferation. Although this paper addresses the early responses of melanocytes, it would be valuable to state whether those responses persist or not.

We sincerely appreciate the reviewers’ comment regarding the concept of short-term and long-term / “durable” effects of BRAF expression. We do continue to monitor cell growth in TPA free media both with and without Dox, and the phenotype continues for the duration of the culture, which we now mention in line 352. In full transparency, “duration” is usually ~4-6 weeks – the primary human melanocytes we use are not immortalized and do crash due to telomere crisis. The time until crash is dependent on the prep and the rate of divisions, regardless of BRAF status. We are presently conducting studies repeating these experiments on the background of different combinations of melanoma-associated mutations, inclusive of TERT promoter and CDKN2A loss mutations, but these studies are the foundation of an independent project in the lab and are outside the scope of this manuscript.

3. Several independent results in the paper support the view that the proliferation of non-BRAFV600E-expressing melanocytes doesn't depend at all on AURKB function, which is surprising given the key role attributed to Aurkb in mitosis. Do the authors think there is an alternate pathway of cell cycle control operating in these cells? Or do they think it's more of a quantitative matter, e.g. that modest, low-level proliferation can be sustained without (much) AURKB but a more rapid, aggressive proliferative state requires it?

Pathways of cell cycle control alternative to AURKB have been identified, but from our transcriptional analyses, we see no evidence of them here. We suspect the most likely reason for the differences in sensitivity is the aneuploidy that results from BRAFV600E. Two recent Nature papers (https://pubmed.ncbi.nlm.nih.gov/33505028/ https://pubmed.ncbi.nlm.nih.gov/33505027/) demonstrate that whole genome doubling renders cells sensitive to mitotic checkpoint inhibition. Our demonstration that BRAF induces aneuploidy that these cells are more sensitive to AURKB inhibition mirror these findings. It is not that diploid cells are not “at all” responsive to mitotic checkpoint inhibition, they are just significantly less sensitive – the same as we see here. The mechanisms aren’t entirely known, and dissecting the mechanisms will require multiple studies all of their own, but the explanation I have been offered by experts in the field is that the more chromosomes there are that need separated, the more AURKB molecules are required for successful separation. BRAF increases the demand for AURKB but the miRNAs prevent increased AURKB from being produced.

4. The limited sample sizes (n = 3) in earlier experiments in addition to the lack of in vivo work lessens the potential clinical impact this manuscript could have on the melanoma community. If it is possible to extend the sample size for some of this work, it would improve the importance of the study.

We thank the reviewers for this feedback. We have guessed that the earlier experiments being referred to are those in Figure 1 demonstrating growth arrest induced by miRNA overexpression. As requested, we have now extended this sample size by repeating over-expression experiments with additional melanocyte preps (Figure 1J). We also point out that Figures 1D, E, G and J each report on reduced proliferation and each have 3 replicates, and in addition the screen represented in Figure 2C included multiple wells of each miRNA as controls and was likewise thrice repeated. In short, the phenotype is very reproducible.

We are sorry to hear the reviewers are skeptical of the long-term clinical impact of this study due to lack of in vivo model systems. Although we respect this viewpoint, we are a bit more optimistic. It is our lab philosophy that there is value in staying as close to the “human model system” as possible. Like other of our studies, this study first generates hypotheses through observation of clinical specimens, then explores those hypotheses in primary human melanocyte culture, then confirms the major mechanisms using clinical specimens again. Will this approach yield observations that are ultimately clinically impactful? We hope to know the answer to that soon enough. The data presented here have resulted in funding for pre-clinical and clinical studies to test the utility of our observations for chemoprevention or early detection of melanoma. We hope to report on these studies in a few years. In the meantime, there are many other fantastic labs that utilize elegant model animal systems, and we have been excited to see some of them making discoveries in their systems that support some of the observations we have made here. We suspect that human-focused studies, such as ours, that complement animal models and determine which observations are conserved in human and which are model-specific will be ultimately useful for the field.

In lines 741-751 of the discussion, we now better acknowledge these merits and weaknesses as well as published studies in animal models that support our observations:

“While supported by analyses of both primary human melanocytes and clinical specimens, one limitation to this report is the lack of in vivo confirmation using animal models. One well-established genetically engineered mouse model of melanoma expresses BRAF^V600E^ from the endogenous locus in tyrosinase expressing cells with spatiotemporal control. The predominant phenotype of BRAF^V600E^ activation in this model is melanocytic hyperplasia, which increases in prominence with BRAF^V600E^ dose^6^. While growth-arrested melanocytic “nevus-like” lesions also form in this model, they are not associated with markers of senescence^8^. More recently, an elegant study utilizing a zebrafish model of melanoma demonstrated that driving BRAF^V600E^ expression from promoters associated with melanocyte progenitors, but not differentiated melanocytes, resulted in melanoma initiation^61^. These observations are all consistent with our data and support the conclusion that BRAF^V600E^-induced proliferation in melanocytes is dependent on differentiation state.”

5. Some very specific questions that the authors should address are the following:Figure 2E: the detected miRNA binding sites in AURKB and GPR3 seem a little weak; how do these sequences rank among other detected targets for these miRNAs in the genome?

MicroRNAs bind mRNAs with only partial complementarity. The binding sites of MIR211 and MIR328 include the seed sequence of the miRNAs which are the most important sequence for targeting mRNAs. Successful binding is dependent not only on this sequence, but also on the expression of the target gene relative to other target genes and the availability of the binding site. We generally use programs such as TargetScan or miRWalk to identify candidate targets, but knowing how context-dependent miRNA targeting is, we don’t put a lot of stock in the exact ranking. Instead, as we did here, we look for genes that are predicted as targeted and also inversely expressed in the context we care about – in this comparing nevi to melanoma specimens. Figure 2C, y-axis, shows how downregulated the AURKB transcript is upon MIR211 over-expression in comparison to the other predicted targets. By this empirical measurement, it is one of the top most downregulated targets.

Regarding a priori ranking of predicted targets using computational methods, miRWalk is a relatively recent method that incorporates a machine learning based approach, and considers numerous variables in addition to sequence to assign a “probability of binding” score. This tool predicts that MIR328-3p and MIR211-5p have 0.85 and 0.96 probabilities of binding AURKB, respectively, and the two MIR328-3p sites have 1.00 probabilities of binding GPR3. We now include these calculated probabilities in the revised Figure 2.

Figure 3G: Notwithstanding the presentation value of displaying cell number relative to the dox-negative control, doing so suppresses information about how TPA itself affects cell number. Thus, the control levels to which data were normalized should be given, at least in the figure legend.

The control doubling times are now included in the legend as requested.

Figure 4C Data availability: Although the PCA data are shown for four separate conditions: plus-or-minus TPA and plus-or-minus Dox, the underlying mRNA data in Supplemental Table 5 merge pairs of conditions (No Dox vs. DoxAll, and TPAAll vs TPAFreeAll). The authors do present the microRNA data separated into four conditions. They should do the same with the mRNA data.

This was an oversite. Our sincere apologies and also gratitude for bringing it to our attention. The updated Supplementary File 5 contains the pairs of conditions for mRNA.

Figure 5F: The published IC50 for Barasertib for Aurkb is 0.36 nM, but the half-effective dose in Figure 5F seems to be about 12-25 nM. Do the authors have an explanation for why the difference is so large?

The published IC50 of 0.36nM is in reference to a cell-free assay. IC50s referencing cell proliferation assays are not expected to be equal to IC50s referencing the activity of the kinase in a cell-free biochemical assay. The two assays contain substantial differences in potential off-target kinases, where the compound needs to be localized to be effective, and all the factors related cell division that may not be directly related to inhibition of the kinase. For example, the IC50 of the well documented BRAFV600E inhibitor, vemurafenib, in cell free assays is 31nM, but can range from 3-5µM in cell lines (PMID: 25623468). Published IC50s of barasertib in melanoma cell lines (same citation) are in the 30-300nM range. We also emphasize that we report GR50s, which account for differences in the baseline growth rates of the cells – an additional variable that cell-free assays cannot account for.